# Functional dissociation of stimulus intensity encoding and predictive coding of pain in the insula

Stephan Geuter[1,2,3]*, Sabrina Boll[1,4], Falk Eippert[5], Christian Büchel[1]

[1]Department of Systems Neuroscience, University Medical Center Hamburg Eppendorf, Hamburg, Germany; [2]Institute of Cognitive Science, University of Colorado Boulder, Boulder, United States; [3]Department of Psychology and Neuroscience, University of Colorado Boulder, Boulder, United States; [4]Department of General Psychiatry, University Hospital Heidelberg, Heidelberg, Germany; [5]Centre for Functional Magnetic Resonance Imaging of the Brain, University of Oxford, Oxford, United Kingdom

**Abstract** The computational principles by which the brain creates a painful experience from nociception are still unknown. Classic theories suggest that cortical regions either reflect stimulus intensity or additive effects of intensity and expectations, respectively. By contrast, predictive coding theories provide a unified framework explaining how perception is shaped by the integration of beliefs about the world with mismatches resulting from the comparison of these beliefs against sensory input. Using functional magnetic resonance imaging during a probabilistic heat pain paradigm, we investigated which computations underlie pain perception. Skin conductance, pupil dilation, and anterior insula responses to cued pain stimuli strictly followed the response patterns hypothesized by the predictive coding model, whereas posterior insula encoded stimulus intensity. This novel functional dissociation of pain processing within the insula together with previously observed alterations in chronic pain offer a novel interpretation of aberrant pain processing as disturbed weighting of predictions and prediction errors.

*For correspondence: stephan. geuter@colorado.edu

**Competing interests:** The authors declare that no competing interests exist.

## Introduction

Classic bottom-up views construe perception as a feedforward stream of sensory information that is passed along the neural hierarchy from receptors to high-level brain regions (*Hubel and Wiesel, 1959*). Accordingly, neurons are thought of as feature detectors and cortical responses to sensory stimuli are expected to scale with the presence of stimulus features, that is, activity in pain processing brain regions should reflect the activation level of nociceptors. This basic account has been extended by a wealth of findings demonstrating that top-down expectations play an important role in modulating both the pain experience and the activity in pain processing brain regions (*Sawamoto et al., 2000*; *Koyama et al., 2005*; *Lorenz et al., 2005*; *Brown et al., 2008*; *Atlas et al., 2010*; *Bingel et al., 2011*; *Wiech et al., 2014b*). Other neuroimaging studies have shown that stimulus-response functions differ between brain regions (*Davis et al., 1998*; *Coghill et al., 1999*; *Apkarian et al., 2001*; *Bornhövd et al., 2002*; *Davis et al., 2002*; *Porro et al., 2004*) and that brain activation is modulated by concurrent task demands (*Bantick et al., 2002*; *Valet et al., 2004*; *Wiech et al., 2005*; *Seminowicz and Davis, 2007*; *Villemure and Bushnell, 2009*).

However, these theories cannot explain the reduction in sensory cortical activity for expected compared to unexpected stimuli (*Alink et al., 2010*; *Egner et al., 2010*; *Todorovic et al., 2011*;

**eLife digest** All over the human body, there are receptors that help to alert the brain to potential harm. For example, intense heat on the skin elicits a signal that travels to the brain and activates many parts of the brain. Some of the same brain regions that are switched on by signals of potential bodily harm also help the brain to form expectations about events. A person's expectations may have a strong influence on how they experience pain. For example, if a person expects that taking a pill will reduce their pain, they may feel less pain even if the pill is a fake.

Exactly how the brain processes pain signals and expectations remains unclear. Does the brain activity simply reflect how intense the heat is? Some scientists think there may be two separate processes going on: one that predicts what will happen and another that calculates the difference between the prediction and what the receptors actually detect. This difference is called a prediction error. If every unpredicted sensory signal elicits a calculation of the prediction error that would help improve the brain's future predictions.

Now, Geuter et al. show that the predictions are a key part of how the brain perceives pain induced by heat. In the experiments, 28 people had heat applied to skin on their forearm at temperatures that were either noticeable but not painful or painful. Their brain activity was recorded using functional magnetic resonance imaging (fMRI), and measurements were taken of the pupils in their eyes and their skin's response to heat. The fMRI scans showed that activity in the back part of a brain region called the insular cortex reflects the intensity of the heat that is applied to the person's arm, while the front part of the same region signals pain predictions and the prediction error.

This suggests that scientists are correct that pain predictions and prediction error calculations are an integral part of the pain response. More studies are needed to determine if these brain processes might contribute to chronic pain and whether a similar process occurs in response to other types of unpleasant experiences.

*Kok et al., 2012*). In contrast, theories of Bayesian perceptual decision making, as formalized in predictive coding models (*Knill and Pouget, 2004*; *Friston, 2005*; *Summerfield and de Lange, 2014*), can explain such expectation suppression effects. Their proposal is that perception arises from the integration of sensory input with predictions about upcoming stimuli continuously generated by an internal model. More formally, the percept is determined by the posterior probability as computed by Bayes' theorem from the predictions (prior) and the sensory input (likelihood of a given stimulus). Within this framework, measurements of brain activity are composed of the activity of two distinct neuronal populations – one population encoding the expected stimulus based on an internal model of the world (prediction) and one population encoding the mismatch between sensory input and the prediction (prediction error; PE) (*Rao and Ballard, 1999*; *Friston, 2005*).

A direct hypothesis derived from this framework is that sensory brain responses should be reduced when the brain's prediction was accurate. In this situation, the resulting PE is small and regional brain activation is lower for accurate than for inaccurate predictions. This has been observed for primary visual cortex (*Alink et al., 2010*; *Kok et al., 2012*), early auditory electrophysiological responses (*Todorovic et al., 2011*), and the fusiform face area (*Summerfield et al., 2008*; *den Ouden et al., 2010*; *Egner et al., 2010*). The organization of cortical pain processing differs from other sensory modalities in that many cortical pain processing areas receive direct thalamic input and thus avoid a clear hierarchical organization (*Craig, 2002*; *Dum et al., 2009*). It is therefore unclear whether the same computational principles apply to pain as well. If pain processing is also based on predictive coding principles, this framework would offer an elegant and general computational mechanisms of perception across modalities (*Wiech, 2016*) and could help explain several expectation-related effects, including placebo effects (*Petrovic et al., 2010*; *Büchel et al., 2014*; *Tabor et al., 2017*).

In order to arbitrate between possible mechanisms underlying pain perception, we used a probabilistic heat pain task to formally compare a predictive coding model against a stimulus intensity model and a stimulus plus expectation model, respectively (*Figure 1A–C*). Three different visual cues manipulated expectations about an upcoming cutaneous heat stimulus (*Figure 1D*). Each cue

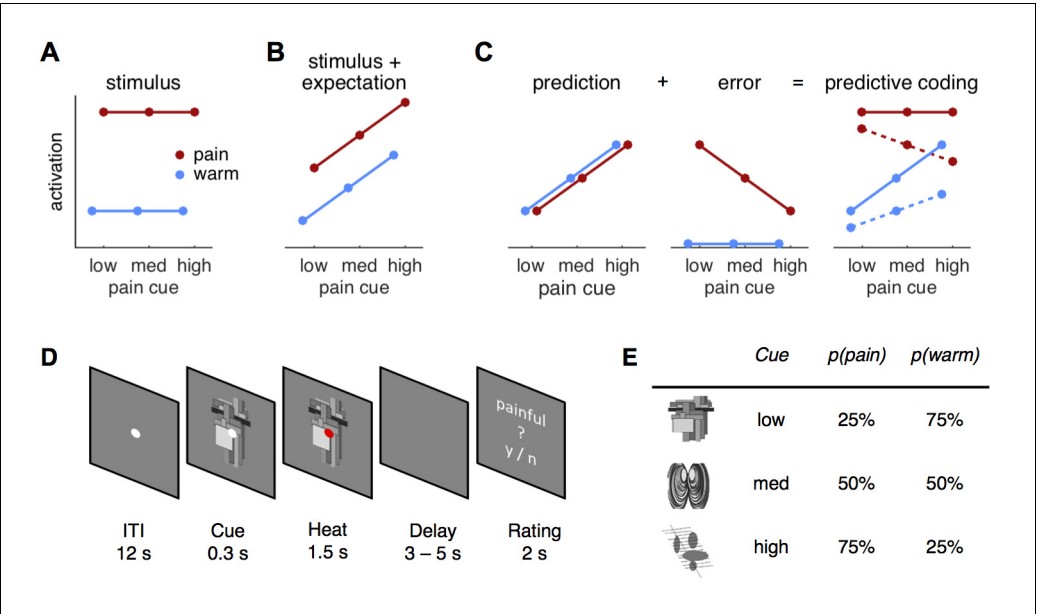

**Figure 1.** Hypotheses and design. (**A**) The stimulus intensity coding model is insensitive to predictive cues and postulates only a main effect of temperature. (**B**) Expectation may have an additive effect on brain responses in that a higher expectation of receiving pain results increased pain and increased physiological responses. (**C**) The predictive coding model has two components; prediction and prediction error (PE). Pain processing regions increase activity with increasing predictions of pain (from low to high pain probability). If the stimulus is painful, a PE signaling the difference between sensory input and the prediction occurs. In accordance with previous studies, we modeled the error for warm stimuli as zero (see Materials and methods, Results). The hypothesized predictive coding response is a weighted sum of the two components. The model has two free weight parameters; both are required to be positive. Solid lines represent equal weighting, while dashed lines represent a higher weighting for the PE. (**D**) Subjects saw a central fixation dot during a 12 s inter-trial-interval (ITI). A cue indicating the probability of a painful stimulus in the current trial appeared 300 ms before the heat stimulus started. Duration of heat stimulation was 1.5 s during which the cue was still visible. After a variable delay of 3–5 s, a rating screen appeared for 2 s and subjects reported whether the last stimulus had been painful or not. The fixation dot changed its color in 12.5% of the trials and participants responded to this change with a button press. (**E**) Cues predicted pain with 25, 50, or 75% probability and were counterbalanced across subjects.

was associated with a different probability of receiving painful or non-painful heat on the forearm (25, 50, or 75% chance of receiving pain and referred to as low, medium, and high cue, respectively; *Figure 1E*). Using functional magnetic resonance imaging (fMRI) in combination with model-based analyses in this task, we quantified evidence for all models in skin conductance responses (SCR), pupil diameter, and across the brain.

## Results

According to the predictive coding model (*Egner et al., 2010*; *Büchel et al., 2014*), responses to cutaneous heat are the weighted sum of the prediction and the PE (*Figure 1C*). Under this framework, pain-processing regions will increase their activity with increasing probability of pain. Furthermore, they will also signal a PE if the stimulation is more painful than expected, but are not expected to show a PE for warm stimulation (*Figure 1C*). This pain PE is motivated by previous work (*Egner et al., 2010*; *Büchel et al., 2014*; *Summerfield and de Lange, 2014*), by the observation that PE for warm stimuli have topographies distinct from PE for pain (*Ploghaus et al., 2000*; *Zeidan et al., 2015*), and findings suggesting that reward and aversive PE are encoded by different neuronal populations (*Yacubian et al., 2006*; *Belova et al., 2007*; *Seymour et al., 2007*; *Fiorillo, 2013*). In addition to the pain PE, we later consider models using absolute and signed PEs, respectively (see *Comparing different PE types*).

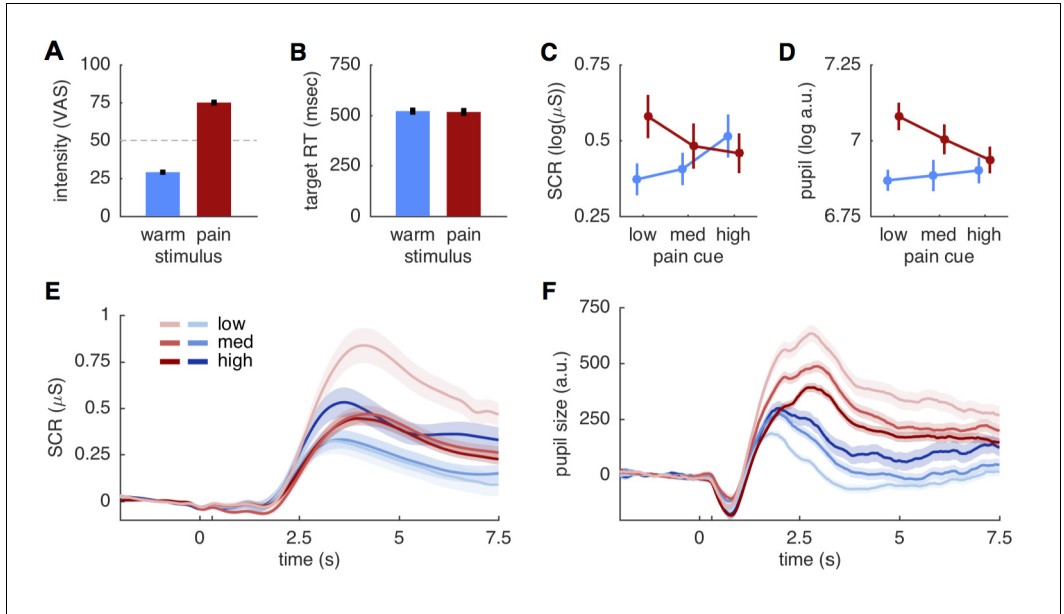

**Figure 2.** Behavioral and physiological results. (**A**) Intensity ratings reported at the end of each block for warm and painful stimuli, respectively. Intensity ratings were significantly higher for pain stimuli ($t(27) = 20.9$; p<0.001) and correspond well to the stimulation levels chosen during calibration (30 and 75). 'Pain threshold' was marked at the center (50) of the visual analogue scale (VAS) used for these ratings. Error bars in all plots show the standard error of the mean. (**B**) Target reaction time did not differ between stimulation intensities ($t(27)=0.51$; p=0.61). (**C**) Skin conductance responses (SCR) for pain (red) and warm (blue) stimuli. SCR responses reflect the pattern hypothesized by the predictive coding model. (**D**) Pupil dilation amplitudes shows the same response pattern as SCR, also supporting the predictive coding model. (**E**) Evoked skin conductance responses (SCR) for warm (blue) and painful (red) stimuli are plotted for each condition and followed the rank order hypothesized by the predictive coding model. (**F**) Pupil diameter responses plotted using the same groupings as in (**D**). SCR and pupil traces are aligned to cue onset at 0 s, stimulus onset is at 300 ms (unlabeled tick mark), and shaded areas indicate standard errors.

Alternatively, pain-processing regions could simply encode stimulus intensity (*Figure 1A*) or an additive combination of intensity and expectation (*Figure 1B*). Each model makes different predictions about the measured response profiles within the present paradigm, which we tested both in an analysis of variance (ANOVA) framework and using formal Bayesian model comparison.

## Behavioral results

Before comparing the different models against each other, we verified that the two stimuli were clearly distinguishable. Pain ratings obtained after each run showed that the 28 participants distinguished between the two stimulus intensities ($t(27) = 20.9$; p<0.001), that intensity ratings were close to the calibrated intensities of 30 and 75, respectively (mean warm: $29.0 \pm 9.1$ std.; mean pain: $75.0 \pm 10.3$ std.), and that warm stimuli were not perceived as painful (*Figure 2A*). Trial-by-trial ratings classifying stimuli as either painful or non-painful matched the stimulus intensity with 94.3% accuracy, further supporting the qualitative difference between the two stimuli. Target reaction times to color changes of the fixation dot did also not differ between two stimulus intensities (warm $520.3 \pm 94$ ms; pain $516.1 \pm 104.8$ ms; $t(27)=0.51$; p=0.61; *Figure 2B*), suggesting a similar attention allocation for both stimulation intensities.

## Skin conductance and pupil responses

From an ANOVA perspective, the stimulus intensity model predicts a main effect of stimulus, whereas the stimulus plus expectation model predicts an additional main effect of cue (*Figure 1A, B*). By contrast, the summation of predictions and PE in the predictive coding model should result in a cue × stimulus interaction (*Figure 1C*). We thus computed ANOVA's for SCR, pupil dilation and

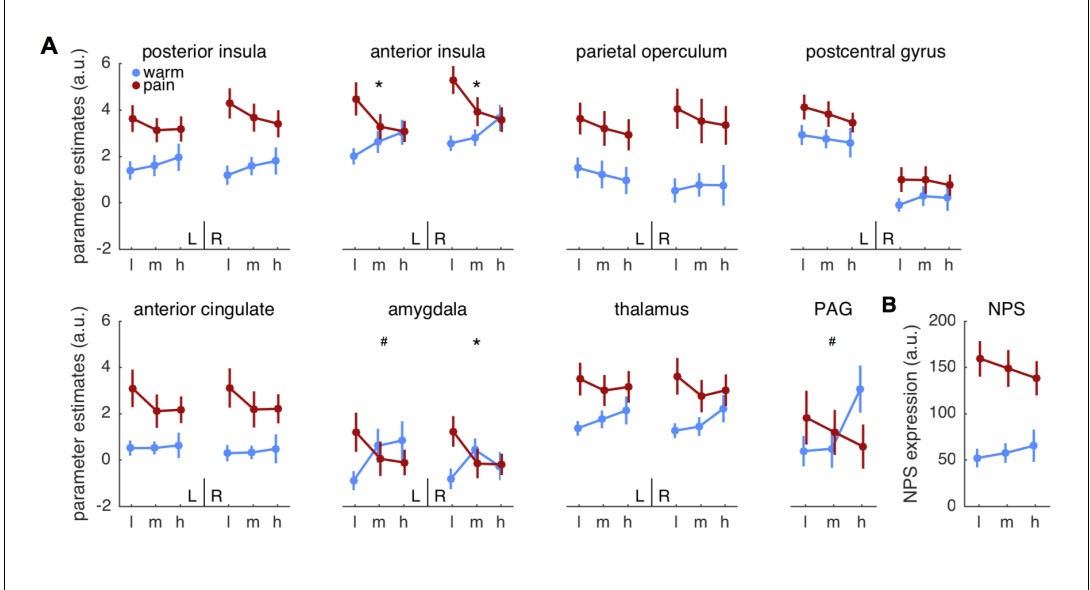

**Figure 3.** Parameter estimates for regions of interest. (**A**) Mean parameter estimates (± standard error) are plotted for left (L) and right (R) hemispheres in each panel, except for the midline structure PAG. Blue indicates warm stimuli, red indicates painful stimuli. Cues are on the x-axis, with 'l' designating low, 'm' designating medium, and 'h' designating high probability of pain. PAG = periaqueductal gray. (**B**) Pattern expression for the neurological pain signature (NPS; *Wager et al., 2013*). *interaction effect significant FDR corrected $q < 0.05$. #interaction: $p<0.05$, uncorrected.

brain data before conducting formal model comparisons. Peak amplitudes of both SCR and pupil dilation showed the expected interaction effect (SCR: $F_{(2,40)}=27.7$; $p<0.001$; pupil: $F_{(2,38)}= 9.5$; $p<0.001$). Responses of both measures increased with higher probability for pain when the stimulus

**Table 1.** ANOVA results for brain ROI and NPS.

| Region | | Stimulus $F_{(1,27)}$ | P | Cue $F_{(2,54)}$ | P | Cue X stimulus $F_{(2,54)}$ | P |
|---|---|---|---|---|---|---|---|
| ACC | L | 13.93 | 0.0009* | 1.21 | 0.3053 | 1.23 | 0.3017 |
| | R | 15.99 | 0.0004* | 0.95 | 0.3923 | 1.11 | 0.3372 |
| anterior insula | L | 8.3 | 0.0077* | 0.41 | 0.6651 | 5.46 | 0.0069* |
| | R | 9.69 | 0.0043* | 1.58 | 0.2155 | 7.48 | 0.0014* |
| posterior insula | L | 15.73 | 0.0005* | 0.28 | 0.7538 | 1.58 | 0.2145 |
| | R | 12.12 | 0.0017* | 0.15 | 0.8637 | 0.09 | 0.9111 |
| parietal operculum | L | 18.3 | 0.0002* | 1.2 | 0.3089 | 0.02 | 0.9779 |
| | R | 23.35 | <0.0001* | 0.17 | 0.8408 | 0.72 | 0.4918 |
| post central gyrus | L | 6.14 | 0.0198 | 1.1 | 0.3409 | 0.18 | 0.839 |
| | R | 2.57 | 0.1206 | 0.18 | 0.8387 | 0.41 | 0.6675 |
| amygdala | L | 0.1 | 0.7506 | 0.1 | 0.9046 | 4.94 | 0.0107 |
| | R | 0.83 | 0.369 | 0.51 | 0.6033 | 5.39 | 0.0074* |
| thalamus | L | 8 | 0.0087* | 0.39 | 0.6761 | 1.4 | 0.2545 |
| | R | 7.6 | 0.0104* | 1.32 | 0.275 | 2.62 | 0.0823 |
| PAG | | 0.02 | 0.9027 | 1.02 | 0.3675 | 4.34 | 0.0178 |
| NPS | | 47.73 | <0.0001* | 0.14 | 0.8708 | 2.18 | 0.1228 |

ACC: anterior cingulate cortex, PAG: periaqueductal gray, NPS: neurological pain signature.
*FDR $q<0.05$.

was non-painful, but responses were lower expected pain compared to unexpected pain (*Figure 2C, D*). This response profile mirrors the profile hypothesized by the predictive coding model (*Figure 1C*). Plotting the grand means of the evoked SCR and pupil responses confirmed the rank-order of conditions observed in the peak amplitude analyses (*Figure 2E,F*).

In addition to the interaction effects, the main effect of stimulus was also significant for SCR ($F(1,20)=7.5$; $p=0.012$) and pupil dilation ($F(1,19)=32.5$; $p<0.001$). In both cases the overall response was stronger for the painful than for the non-painful stimuli (*Figure 2C,D*). The main effect of cue was also significant for the SCR ($F(2,40)=4.6$; $p=0.015$), but was not significant for the pupil dilation ($F(2,38)=2.7$; $p=0.078$). Hence, the ANOVA results are compatible with both the predictive coding and stimulus intensity model, while the SCR cue effect is also predicted by the stimulus plus expectation model. However, a post-hoc t-test comparing SCRs to painful and warm stimuli following a high cue did not reveal the difference proposed by the stimulus plus expectation model ($t(20)=1.54$; $p=0.14$; *Figures 1B* and *2C*).

## Region of interest results

We next computed ANOVA's on brain activity extracted from anatomically defined *a priori* regions of interest (ROI). Among those ROIs, bilateral anterior insula (left: $F(2,58)=5.5$; $p=0.007$; right: $F(2,58)=7.5$; $p=0.001$) and right amygdala ($F(2,58)=5.4$; $p=0.007$) showed the expected cue × stimulus interaction (*Figure 3A*, *Table 1*). Importantly, the response pattern matched the pattern expected by the predictive coding model, that is, responses in anterior insula and amygdala increased with pain expectation for warm stimuli and decreased for pain stimuli. Furthermore, all regions except for the postcentral gyrus, amygdala and PAG showed a significant main effect of stimulus (*Table 1*), but no ROI showed a significant main effect of cue.

Although the above ROIs are associated with pain processing, a recently developed multivariate pattern, termed neurological pain signature (NPS; *Wager et al., 2013*), provides a more specific and sensitive estimate of heat pain intensity (*Wager et al., 2013*; *Krishnan et al., 2016*). We thus computed an ANOVA on the pattern expression values as indicators of overall pain intensity for the NPS (*Figure 3B*). Stimulus intensity had an effect on NPS expression ($F(1,27)=47.7$; $p<0.001$), but neither cue nor the interaction were significant (both $p>0.12$; *Table 1*). Since NPS responses are strongly correlated with experimental heat pain reports (*Wager et al., 2013*; *Krishnan et al., 2016*), they can potentially serve as an indicator of trial-by-trial pain reports in this context to test for effects of correct predictions on pain reports. A *post-hoc* t-test revealed that unexpected pain tended to elicit stronger responses than expected pain ($t(27)=2.2$; $p=0.036$).

## Voxel-wise statistical maps

In order to test for the proposed effects in brain regions outside of the *a priori* defined ROIs, we computed a whole brain analysis for the effects of stimulus (pain > warm), effects of cue (cue high > cue low), and for the interaction contrast ((cue high, warm) > (cue low, warm)) > ((cue high, pain) >

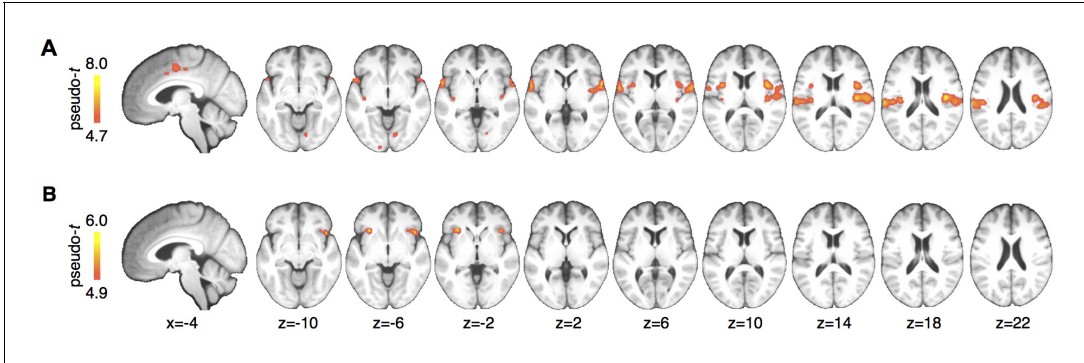

**Figure 4.** Whole brain results. (**A**) A main effect of stimulus was observed in pain processing regions including insula, parietal operculum, and midcingulate cortex. (**B**) Anterior insula showed a significant interaction between cue and stimulus. Maps are displayed at p<0.05, whole brain FWE corrected using nonparametric permutation testing resulting in pseudo-t maps.

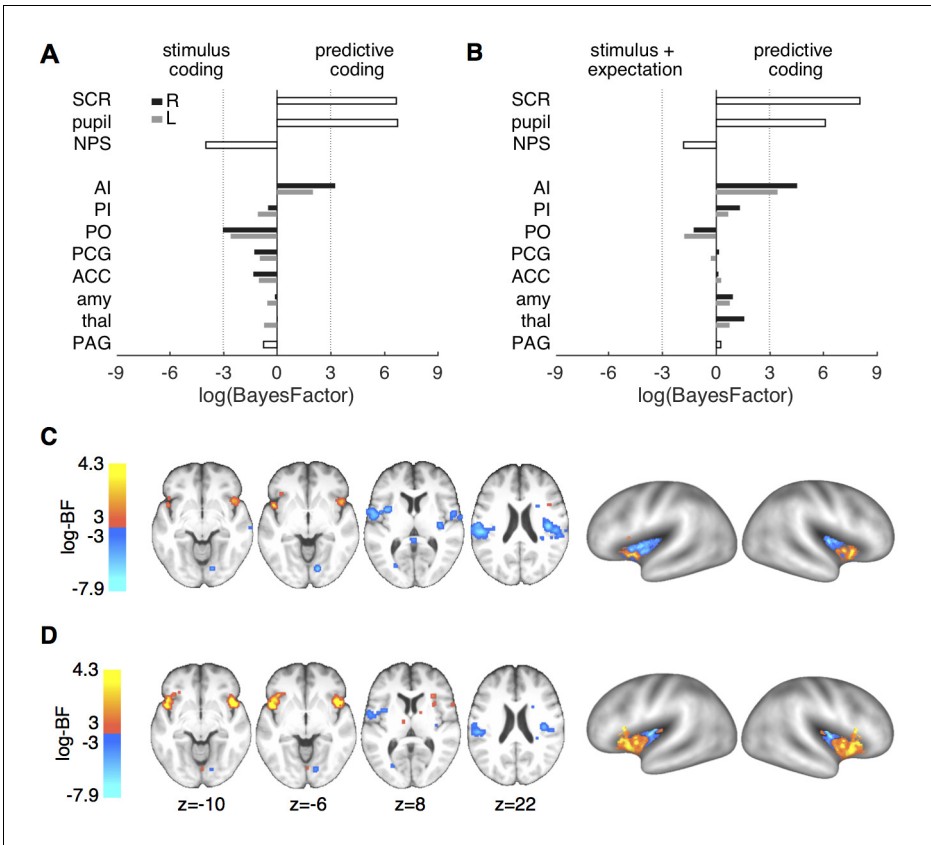

**Figure 5.** Formal model comparison. (A) log-BF comparing the predictive coding model against the stimulus intensity model for SCR, pupil, NPS, and ROIs. SCR, pupil and right anterior insula show strong evidence for predictive coding (*log-BF >3*), while NPS and posterior insula favor the stimulus intensity model (*log-BF < −3*). (B) log-BF comparing the predictive coding model against the stimulus plus expectation model. Results are similar to (A), but evidence for the stimulus plus expectation model is weaker. (C) Voxel-wise log-BF comparing the predictive coding model against the stimulus intensity model and in (D) against the stimulus plus expectation model. Maps are thresholded at |*log-BF*|>3. Warm colors indicate support for the predictive coding model, cold colors indicate support for the alternative model. Surface projections of unthresholded log-BF insula maps reveal an anterior-posterior gradient. AI, anterior insula; PI, posterior insula; PO, parietal operculum; PCG, post-central gyrus; ACC, anterior cingulate cortex; amy, amygdala; thal, thalamus; PAG, periaqueductal gray.

(cue low, pain)). The stimulus intensity contrast revealed activations in classical pain processing areas, including posterior and mid-insula, secondary somatosensory cortex, parietal operculum, and mid-cingulate cortex (*Figure 4A*). A cue × stimulus interaction was observed again in left (peak MNI coordinates: x=−30, y = 24, z=-4) and right (x = 46, y = 20, z=−8) anterior insula (*Figure 4B*). No other brain region showed the interaction effect at a family wise error rate of p<0.05. Testing for the main effect of cue did not reveal any significant voxels.

## Formal model comparisons

After observing that the response profiles of SCR, pupil, bilateral anterior insula, and right amygdala were as expected by a concurrent representation of predictions and PE, we conducted formal model comparisons using Bayes factors (BF) (*Kass and Raftery, 1995*; *Rouder and Morey, 2012*) to identify the best explanatory model. Bayes factors are computed as the ratio of marginal likelihoods of the data under each of two models and thus quantify the evidence for one model over the other given the data. This metric thus allows the identification of the best model while implicitly controlling for the number of free parameters. Following *Kass and Raftery (1995)*, we consider *log-BF >3* as strong evidence for the predictive coding model and values of *log-BF < −3* as strong evidence for the alternative model.

Comparing the predictive coding model against the stimulus intensity model revealed strong evidence in favor of the predictive coding model for both SCR and pupil responses (log-BF$_{SCR}$ = 6.65; log-BF$_{pupil}$ = 6.75; *Figure 5A*). Comparing it against the stimulus plus expectation model revealed similarly decisive evidence in favor of the predictive coding model (log-BF$_{SCR}$ = 8.05; log-BF$_{pupil}$ = 6.09; *Figure 5B*). These log-BF values indicate that the predictive coding model was at least 400 times more likely than each of the two alternatives.

In contrast, NPS expression was better explained by the stimulus intensity model compared to the predictive coding model (log-BF = −3.98), mirroring the previously observed main effect of stimulus (*Figure 5A*). Computing log-BF's for the individual ROIs confirmed the results of the ANOVA interaction tests in that the anterior insula showed strong evidence for the predictive coding model compared to the two alternative models (*Figure 5*). Generally, the right hemisphere yielded a clearer picture in terms of model evidence, potentially because of stronger signal in the hemisphere contralateral to the heat stimulation on the left arm. For example, evidence for the predictive coding model against the stimulus intensity model in the left anterior insula ROI was below threshold, while the evidence was above threshold for the right anterior insula (*Figure 5A,B*). Interestingly, the right parietal operculum ROI showed strong evidence for the stimulus intensity model (log-BF = −3.02; *Figure 5A*).

Comparing the stimulus intensity against the stimulus plus expectation model did not reveal decisive evidence for one over the other model on the ROI level. Although no comparison reached the threshold of |log-BF|>3, all ROIs and physiological measures weakly supported the simpler, stimulus intensity model (log-BF range: 0.12–2.16).

In order to obtain a spatially more detailed picture of the computational processes of pain processing across the brain, we computed voxel-wise log-BF's comparing the predictive coding model against the stimulus intensity model and the stimulus plus expectation model, respectively. Again, responses in bilateral anterior insula strongly supported the predictive coding model (*Figure 5C,D*). Within the posterior insula and parietal operculum, this more fine-grained analyses revealed bilateral evidence for the simpler, stimulus-intensity model, which was less evident on the ROI level. Similar results were obtained when comparing the predictive coding model against the stimulus plus expectation model (*Figure 5D*), but evidence for the stimulus plus expectation model was weaker. Directly comparing the stimulus intensity model against the stimulus plus expectation model revealed modest support for the stimulus intensity model (*log-BF >2*) in midcingulate cortex, posterior insula, and parietal operculum.

A surface projection of the non-thresholded, voxel-wise log-BF maps comparing the predictive coding model against the two alternative models within the insula, demonstrated a gradual change in evidence from anterior to posterior insula (*Figure 5C,D*). This gradient is also evident when the average log-BF from the insula is plotted over the anterior-posterior dimension (*Figure 6A*). Importantly, and in line with anatomical considerations, the gradient is also steeper in the right hemisphere

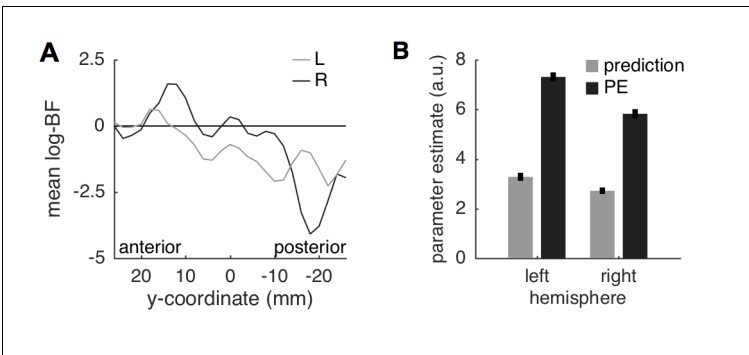

**Figure 6.** Insula results. (A) Plotting the average log-BF for left and right insula against y-coordinates shows the anterior-posterior gradient from predictive coding to stimulus intensity coding. (B) Weight parameters are positive for prediction and PE terms, as postulated. The PE contributes approximately two times as much to the anterior insula signal as the prediction does.

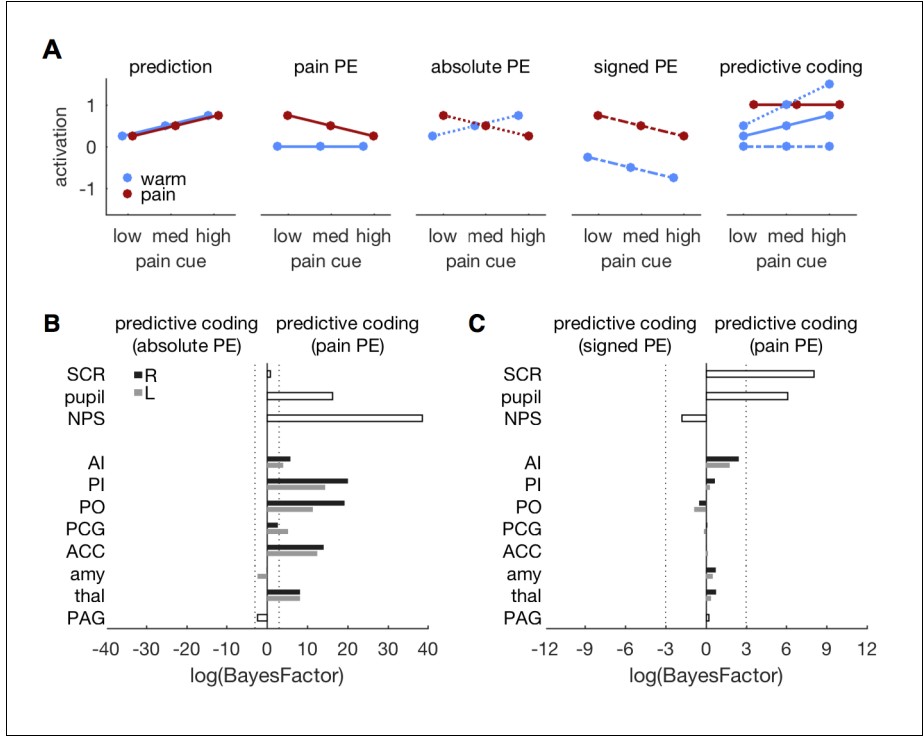

**Figure 7.** Comparing different PE types. (**A**) Different variants of the predictive coding model. All variants share the same prediction term (as in *Figure 1A*), but differ in the computation of the PE. The original model used here, specifies a pain PE, which equals zero for warm stimuli (second panel, solid lines). An alternative model specifies an absolute PE (third panel, dotted lines). The third alternative model uses a signed PE (fourth panel, dash-dot lines). Please note that all three alternatives result in the same PE for painful stimuli. They only differ in the PE for warm stimuli. The right-most panel shows the expected response profile for each of the three PE definitions when prediction and PE are equally weighted, i.e. simple sum of both terms. Please note that the signed PE (dash-dot line) model does not capture any factorial interactions between cues and stimuli. (**B**) log-BF comparing the absolute PE model against the original, pain PE model for ROIs and autonomic measures. No evidence stronger than *log-BF < −3* is available for the absolute PE model. (**C**) log-BF comparing the signed PE model against the pain PE model shows no decisive evidence for the signed PE model.

(contralateral to stimulation) based on the more decisive evidence in both anterior and posterior insula.

Since we expect both predictions and PEs to contribute positively to the measured brain signal, we extracted the weight parameters for predictions and PEs from the left and right anterior insula regions in which *log-BF >3*. Firstly, the weights for both predictions and PEs were positive as postulated by the model. Interestingly, the weight for the PE was approximately two-times as strong as the prediction (left anterior insula: 1:2.1; right: 1:2.2), which is very similar to the ratio of 1:2 reported in a previous study in the fusiform face area (*Egner et al., 2010*). A stronger weighting of the PE results in reduced responses for expected compared to unexpected painful stimuli as illustrated in *Figure 1C* and observed here in SCR (*Figure 2C*), pupil dilation (*Figure 2D*), and anterior insula and amygdala activation (*Figure 3A*) as well as in other studies (*Alink et al., 2010*; *Todorovic et al., 2011*).

## Comparing different PE types

While neuronal coding of reward prediction errors is well understood (*Schultz et al., 2015*), the specifics of aversive PE coding are currently debated (*Belova et al., 2007*; *Seymour et al., 2007*; *Boll et al., 2013*; *Fiorillo, 2013*; *Klavir et al., 2013*; *Roy et al., 2014*; *Matsumoto et al., 2016*). We therefore compared three versions of the predictive coding model that differed in their PE specification. The original model, presented above, builds on a pain PE in which a warm stimulus does not

elicit a PE within pain regions (*Figures 1A* and *7A*). Alternative models incorporated an absolute PE (i.e., the absolute difference between stimulus and prediction) and a signed PE (i.e., the difference between stimulus intensity and prediction), respectively (*Figure 7A*). All models share the same prediction term.

The original, pain PE model provided a better fit of the pupil, NPS, and ROI responses when compared against the absolute PE model (*Figure 7B*). Only the PAG and bilateral amygdala tended to favor the absolute PE model, but the model evidence did not pass the threshold of *log-BF < −3*. The pain PE model also provided a better fit than the signed PE model for SCR and pupil diameter (*Figure 7C*). For the ROIs, the pain PE also provided better fits, while the signed PE tended to fit the NPS response slightly better. But again, none of the ROI comparisons revealed above threshold evidence. In summary, neither the absolute nor the signed PE model provided compelling evidence for a better fit than the pain PE model. In fact, the pain model explained the responses better than the alternatives in most of the ROIs.

## Discussion

Combining a probabilistic heat pain paradigm and Bayesian model comparison, we identified a functional dissociation between anterior and posterior insula suggesting that these regions implement different computations supporting pain perception. While posterior insula and parietal operculum employ stimulus intensity coding, activity in the anterior insula reflects the summation of pain expectation and prediction errors resulting from unexpected pain, thus conforming to a predictive coding account (*Egner et al., 2010*; *Büchel et al., 2014*). This functional dissociation was evident at a ROI level as well as on a voxel-wise analysis. The combination of expectation and prediction error was also by far the best model explaining SCR and pupil diameter responses to painful and non-painful heat stimuli. By contrast, the response profile of a multivariate brain pattern predictive of pain ratings (NPS; *Wager et al., 2013*) reflected only stimulus intensity.

The insula as a whole is involved in a multitude of different processes – at least 20 according to one review (*Nieuwenhuys, 2012*) – from somatosensory to emotional and conflict processing (*Kurth et al., 2010*; *Chang et al., 2013*; *Wiech et al., 2014a*). Interestingly, the cytoarchitectonic organization of the insula and its functional connectivity with other brain regions vary smoothly along an anterior-posterior gradient (*Cerliani et al., 2012*; *Nieuwenhuys, 2012*). A similar functional gradient was evident in the present study, thereby linking the underlying cytoarchitectonic and connectivity gradients to distinct pain processing modes.

The posterior insula is strongly involved in somatosensory and pain perception (*Mazzola et al., 2012*; *Nieuwenhuys, 2012*), it receives direct spinothalamic input (*Craig, 2002*; *Dum et al., 2009*), and it is functionally and structurally connected to somatosensory cortices (*Wiech et al., 2014a*). This connectivity pattern is well suited to support a stimulus intensity coding role of the posterior insula as observed here using phasic pain stimuli and as also indicated by a more tonic model of pain (*Segerdahl et al., 2015*). It should be noted that the sensory role of the posterior insula includes non-painful stimuli, too (*Davis et al., 1998*).

By contrast anterior insula activity represents a combination of pain prediction and PE. Its predictive function fits nicely with previous reports that pre-stimulus activity in the anterior insula modulates the perception of subsequent stimuli (*Ploner et al., 2010*; *Wiech et al., 2010*). Furthermore, the anterior insula flexibly connects to emotional and attentional brain regions (*Taylor et al., 2009*; *Ploner et al., 2010*) and integrates information from a diverse set of prefrontal and limbic brain regions (*Critchley, 2005*; *Seminowicz and Davis, 2007*; *Kurth et al., 2010*; *Cerliani et al., 2012*; *Wiech et al., 2014a*). In addition to its predictive function, the anterior insula also encodes mismatches between predictions and aversive outcomes during reinforcement learning (*Seymour et al., 2004*; *Pessiglione et al., 2006*; *Boll et al., 2013*) and other concurrent task demands (*Seminowicz and Davis, 2007*). The strong connections between anterior insula and prefrontal regions involved in contextual processing as well as its 'hub-like', evaluative function (*Baliki et al., 2009*; *Chang et al., 2013*; *Uddin et al., 2014*) render the anterior insula particularly suitable for the evaluation of predictions against sensory input. The anterior insula could thus represent a mediator between somatosensory signals in posterior insula and contextual representations in prefrontal cortex, integrating those representations for perceptual decisions and behavioral responses (*Kong et al., 2006*; *Seminowicz and Davis, 2007*; *Baliki et al., 2009*). Integration of multiple

information streams in this brain region could thus be crucial for the construction of pain experiences that are shaped by learning and external feedback (*Wiech, 2016*; *Geuter et al., 2017*).

In addition to the anterior insula, both SCR and pupil diameter in this study – and reaction times in a similar paradigm (*Wiech et al., 2014b*) – showed a pattern predicted by our model. Expectations of certain visual stimuli can also sharpen their cortical representation (*Kok et al., 2012*), but it is unknown how this would translate to pain reports — whether predicted pain would be more or less intense. A recent study did observe no difference in pain ratings between correctly and incorrectly cued pain stimuli (*Zeidan et al., 2015*). Here, we opted for a time-efficient design and did not collect trial-by-trial pain ratings to address this question. However, investigating potential indicators for perceived pain, responses of the NPS (*Wager et al., 2013*; *Krishnan et al., 2016*) and autonomic measures (*Geuter et al., 2014*), can offer insights. All three measures – NPS, SCR, and pupil – showed stronger responses to unexpected compared to expected pain, hinting at a potential enhancement of unexpected pain stimuli that needs to be investigated more thoroughly in futures studies. Taken together, the results show that principles of predictive coding are relevant for behavioral responses in the context of pain.

Other stimulus attributes than painfulness, e.g. general aversiveness, salience, or motivational demands, co-vary with stimulus intensity. Previous studies have correlated brain activity with pain reports and stimulus intensity in order to dissociate the two (*Coghill et al., 1999*; *Büchel et al., 2002*; *Davis et al., 2002*, *2004*; *Porro et al., 2004*; *Baliki et al., 2009*). Interestingly, anterior insula activity correlated with perceived heat even in the absence of heat stimulation (*Davis et al., 2004*). Studies by Downar and colleagues (*Downar et al., 2000*, *2003*) also found that anterior insula and anterior cingulate cortex responded to unexpected or novel stimuli. Within a predictive coding framework, the overall response is decomposed into two distinct functional components – a prediction term and PE term – that are key components of learning theory. Interestingly, the anterior insula also showed a prediction error response in the present study in line with previous work (*Downar et al., 2000*, *2002*; *Seymour et al., 2004*; *Boll et al., 2013*). From a psychological perspective, the decomposition is important because directing attention towards expected aversive events (high probability of pain, prediction term) and towards unexpected events (PE) is adaptive. The prediction error is assumed to drive learning (*Rescorla and Wagner, 1972*) and is thus critical for adaption to the environment. By contrast, salience – understood as the difference to preceding sensory events (*Mouraux et al., 2011*) – emerges after stimuli have been processed and the elicited surprise or PE has been computed. Salience can thus be considered a secondary stimulus property resulting from a high PE that in turn can modulate subsequent updating of beliefs. This process is formalized in the Pearce-Hall model of reinforcement learning in which a surprising, salient outcome, affects the learning rate in the next trial (*Pearce and Hall, 1980*; *Boll et al., 2013*; *Atlas et al., 2016*).

Predictive coding theories offer a parsimonious computational implementation of cross-modal, Bayesian perceptual decision making (*Knill and Pouget, 2004*; *Friston, 2005*; *Summerfield and de Lange, 2014*). These accounts can explain several effects within a single framework including extra-classical receptive field effects in visual cortex (*Rao and Ballard, 1999*), repetition suppression (*Summerfield et al., 2008*; *Todorovic et al., 2011*), and have been suggested as a framework to understand placebo effects (*Büchel et al., 2014*). In support of a domain general integration process of expectations and PEs, the ratio of the contributions of both processes to physiological signals observed here, mirrored the ratio previously reported for the fusiform face area in visual perception (*Egner et al., 2010*). Interestingly, in both studies, the PE was weighted stronger than the prediction, which suggests that learning and updating of the internal model are crucial for perception. In addition, the observation that the anterior insula also processes PEs in other modalities (*Downar et al., 2000*; *Iglesias et al., 2013*) hints towards a cross-modal role of the anterior insula within a predictive coding framework.

Another feature of predictive coding models is their hierarchical organization: At each level of the neural hierarchy, predictions and PEs will be computed for the specific features encoded in this region (*Rao and Ballard, 1999*; *Friston, 2005*). For example, early visual and auditory areas process multimodal stimuli under the assumption of independent physical sources and only higher areas form joint representations using Bayesian inference (*Rohe and Noppeney, 2015*). Although anterior insula activity matched the response pattern proposed by the predictive coding model, other brain regions and the NPS followed a stimulus intensity model. This discrepancy could be either due to

inherently distinct computations implemented in those regions or due to the fact that the nature of predictions and PEs changes across regions (*Iglesias et al., 2013*). Because pain is an inherently multi-faceted experience that includes sensory-discriminative, emotional, and motivational components, the present predictive coding model could capture certain aspects of this multi-faceted experience better than others. The computational difference observed between anterior and posterior insula could, at least in part, reflect such functional differences. In fact, studies investigating visual processing within a predictive coding framework also observed regionally restricted effects based on the manipulated stimulus features. For example, activity of the fusiform face area and parahippocampal place area is well described by a predictive coding model, but each region responds selectively to their respective preferred stimuli, i.e. faces and houses (*den Ouden et al., 2010*; *Egner et al., 2010*; *Jiang et al., 2013*). Similarly, expectations of certain low-level visual features such as grating orientation, selectively attenuate primary visual cortex activity, but not activity in higher visual areas (*Alink et al., 2010*; *Kok et al., 2012*). Based on these results, the observed computational differentiation between anterior and posterior insula indicates that both regions process distinct features of painful stimuli and these could be related to different psychological and behavioral outcomes in healthy and patient populations.

The representation of aversive prediction errors in the brain is still not fully understood. Important open questions include whether aversive PE are represented on a continuous dimension along with reward prediction errors and whether particular brain regions represent absolute, signed, or pain PE. Activity reflecting absolute aversive PE has been observed in the amygdala (*Boll et al., 2013*; *McHugh et al., 2014*), while signed aversive PE have been observed in the striatum and PAG (*Seymour et al., 2005*, *2007*; *Roy et al., 2014*; *Zhang et al., 2016*). Within sensory cortices, unexpected omissions of visual or auditory stimuli lead to enhanced activity in auditory or visual areas, reminiscent of absolute prediction errors (*den Ouden et al., 2009*, *2010*; *Todorovic et al., 2011*; *Todorovic and de Lange, 2012*). Comparing models incorporating different PE specifications, showed that the model based on an asymmetric, pain PE explained brain responses in the present study best. Our results thus suggest that PE encoding in the anterior insula differs between situations when the outcome is more pain than expected compared to unexpected pain omissions. The differentiation of PEs observed here is similar to a distinction observed in visual processing: in two studies, activity in the fusiform face area, a face-selective brain region, reflected prediction errors for face stimuli, but not for house stimuli (*den Ouden et al., 2010*; *Egner et al., 2010*).

In summary, the observed responses in SCR, pupil dilation, and anterior insula activation demonstrate that at least part of the pain experience can be explained by a domain general predictive coding framework. The parallels observed between pain and visual processing (*Egner et al., 2010*) hint towards a general processing principle based on internal predictions and PE. An interesting question for future research is how the contributions of predictions and PE shift in states of altered or chronic pain conditions that are also related to altered learning processes (*Vlaeyen, 2015*). As anterior insula structure and function changes profoundly in chronic pain conditions (*Bushnell et al., 2013*; *Ceko et al., 2013*; *Hong et al., 2014*; *Flodin et al., 2015*), it is possible that the precision or influence of the prediction is strongly enhanced in chronic pain conditions or that PEs are incorrectly computed (*Edwards et al., 2012*). If the underlying computations are domain general, this would also explain the hyper-sensitivity observed in certain chronic pain populations to non-painful tactile and visual stimuli (*López-Solà et al., 2014*). This framework could hence open up new ways to investigate pain processing in clinical populations.

## Materials and methods

### Sample

Twenty-eight healthy subjects (17 female) with an average age of 25.9 years (range: 21–33 years) participated in this study. No subject reported any psychiatric, neurological, dermatological, or pain conditions. Due to equipment malfunction, skin conductance data from seven subjects could not be analyzed (resulting in a sample size of N = 21 for SCR analyses) and technical issues prohibited pupil data collection for eight subjects (leaving N = 20 for pupil analyses); only one participant had neither SCR nor pupil data. Other behavioral and fMRI data analyses are based on the full sample of 28 participants. The sample size was determined as 1.5 times the sample of a seminal fMRI study on pain

expectations that tested 19 subjects (*Atlas et al., 2010*). The Ethics committee of the Medical Chamber Hamburg approved the study.

## Procedure

After arrival at the laboratory, subjects were informed about the procedures of the experiment and provided written informed consent. The experiment was divided into three parts – a temperature calibration phase, a behavioral training session, and the functional magnetic resonance imaging (fMRI) experiment.

First, we calibrated the temperatures to be used in the experiment individually for each subject (outside of the MR-scanner). For calibration, subjects rated 36 cutaneous heat stimuli (total duration: 1.5 s, ramp-up: 70°C/s, ramp-down: 40°C/s) with temperatures ranging from 42°C to 49.5°C (in steps of 0.5°C) in a pseudo-randomized order using a computerized visual analogue scale (VAS). Sixteen different temperatures between 42°C and 49.5°C in steps of 0.5°C were presented two times each during calibration (except for 44, 45, 46, and 47°C, which were repeated three times each), resulting in a total of 36 stimuli. The stimulus interval was 13–17 s plus the time participants needed for their VAS rating (mean: 5.04 s, standard deviation: 1.01 s). Heat stimuli were applied to the left volar fore-arm and different skin sites were used for calibration, behavioral training and fMRI scanning. The extremes of the VAS were labeled 'no sensation at all' and 'unbearable pain'. The center of the VAS was labeled 'pain threshold'. This VAS partition was necessary because we needed to determine one painful and one non-painful, but clearly noticeable level of stimulation for the main experiment. Subjects were instructed to only rate stimuli as above the pain threshold if the stimulus induced any painful sensation. For stimuli that were perceived as different from baseline but not painful, subjects rated the intensity of the warmth on the lower half of the VAS. 'Unbearable pain' was explained to the subjects as the intensity at which they would have to lift the thermode from the arm. VAS ratings were converted to numerical values ranging from 0 to 100. Intensity ratings did not differ between men and women ($t(26) = 1.32$; $p=0.2$). The average correlation across subjects between temperature and rating was high: $\bar{r} = 0.78$ (standard deviation: 0.13). We used linear regression to determine one temperature that was clearly noticed by the subject but not painful (VAS 30) and a second temperature that was perceived as painful but tolerable (VAS 75). We next applied the selected temperatures to the subjects' forearm to ensure that the warm stimulus was not painful, but clearly distinguishable from baseline and that the painful stimulus was bearable – this was the case for every subject. The average temperature for the warm stimulus was 45.0°C (standard deviation: 1.2°C) and the average temperature for the painful stimulus was 49.4°C (standard deviation: 1.3°C) with a maximum temperature of 49.5°C.

Following calibration, subjects were informed about the cues and the contingencies between cues and heat stimuli (*Figure 1E*). The explicit information and the training block ensured that subjects knew the contingencies. The training also minimized learning taking place during the fMRI session. Cue-intensity contingencies were counterbalanced across subjects and subjects were shown their respective pairings on a computer screen. The behavioral training session consisted of one block of 48 trials (see *Task*, below). After the training block, subjects were presented with each of the cues separately on the screen and reported which cue was associated with high, medium, and low probability of pain, respectively. All subjects associated each cue with its correct probability of receiving pain.

After training, subjects were positioned in the MRI scanner and completed 4 blocks of the experiment for a total of 192 trials. The design was identical to the training session, except that each block had a different, pseudo-randomized trial order. The order of blocks was randomized across subjects. The thermode was moved to a different position after each block to prevent sensitization of the skin. During each block, we measured BOLD responses, skin conductance, and pupil diameter. After the end of the fMRI experiment we acquired a high-resolution anatomical image of each subject's head. The whole experiment lasted about 2 h per subject.

## Task

During each trial, a fixation dot was presented centrally on the screen. One of three cues then appeared 300 ms before the heat stimulus started. Heat stimulus duration was 1500 ms (including ~200 ms ramp up and down, respectively). The cue was visible during heat stimulation and

remained on display until the end of the heat stimulation. After a variable interval of 3–5 s, a rating screen appeared asking subjects whether the last stimulus had been painful. Subjects answered 'yes' or 'no' by pressing either the left or right arrow key of a button-box (*Figure 1D*). A fixation dot was presented again during the inter-trial interval (ITI) of 12 s duration. At the end of the training block and after each fMRI block, subjects rated the perceived intensity of the warm and the painful stimuli (on the same VAS as used during calibration). Ratings were in good agreement with the calibrated target ratings of VAS 30 and VAS 75, respectively (*Figure 2A*).

Each cue was presented 16 times in each experimental block. The high pain probability cue was followed by the painful stimulus in 75% of the 16 trials and by the warm stimulus in 25% of the trials. Probabilities for the medium cue were 50% for each stimulus. For the low pain probability cue, the chance for a painful stimulus was 25% and 75% for a warm stimulus (*Figure 1E*). Gray-scale versions of abstract symbols (kindly provided by Dr. Philippe Tobler [*Tobler et al., 2006*]) served as cues (*Figure 1E*).

We included a basic target detection task to ensure that subjects paid attention to the task (*Egner et al., 2010*). In 12.5% of the trials, the fixation dot changed its color to red at the beginning of the somatosensory stimulation. Subjects were asked to respond to the color change by pressing a third key. They were informed that the color change was completely unrelated to the main experimental task. Cues were not related to the color change, as target trials were evenly distributed across cues. During the fMRI experiment, subjects were rewarded with 50 cents for each correct target hit. Detection performance was at ceiling with a minimum of 23 out of 24 correct detections (mean: 23.8). Importantly, the main effect of stimulus on target reaction time was non-significant ($F_{(1,27)}$ = 0.295; p=0.591), indicating that subjects were similarly attentive during pain and warm trials.

## Data acquisition

Stimulus presentation, response logging and thermode triggering were carried out using the Psychophysics Toolbox 3 (http://www.psychtoolbox.org). Thermal stimulation was delivered via a MRI compatible 3 cm diameter Peltier thermode (CHEPS Pathway, Medoc, Israel). Skin conductance was recorded using a Biopac EDA100C MRI system (Biopac Systems, Inc., Goleta, CA, USA) and a CED1401 A/D converter (Cambridge Electronic Design, Cambridge, UK) at a sampling rate of 100 Hz. Electrodes were attached to the thenar and hypothenar eminences of the left hand. Pupil diameter was recorded from the right eye using an MR-compatible EyeLink 1000 system (SR Research, Ottawa, ON, Canada) at a sampling rate of 1000 Hz. The lights in the MRI room were dimmed and luminance was kept constant across subjects. This setup provided a balance between eye-tracking quality and participant comfort.

Functional magnetic resonance imaging (fMRI) data were acquired on a Siemens Trio 3 Tesla system equipped with a 32-channel head coil (Siemens, Erlangen, Germany). Thirty-eight transversal slices (voxel size 2 × 2 × 2 mm, 1 mm inter-slice gap) were acquired within each volume using a T2* sensitive echo planar imaging (EPI) sequence (TR = 2.34 s, TE = 26 ms, flip angle: 80°, field of view: 220 × 220 mm, parallel acceleration factor = 2). Slices were tilted about 30° relative to the AC–PC line to improve coverage in the brainstem. Additionally, T1 weighted structural images (1 × 1 × 1 mm resolution) were obtained using a MPRAGE sequence (TR = 2300 ms, TE = 9 ms, flipangle = 9°).

## Data analyses
### Skin conductance responses (SCR)

The search window for SCRs was constrained to physiologically plausible response onset delays of 1 s or more after cue onset (*Boucsein et al., 2012*), that is, the local minimum at beginning of the SCR had to have a delay of at least 1 s after cue onset and a peak within 10 s after cue onset. We then determined the response amplitudes as the difference between the maximum in the search window the first local minimum. SCR amplitudes were then log-transformed to improve normality before further analyses (*Boucsein et al., 2012*).

## Pupil diameter

Pupil diameter was recorded in epochs of 3 s before stimulus onset to 10 s after stimulus onset to reduce file sizes. Pupil data-recording and analyses followed standard methods previously used (*Einhäuser et al., 2008*; *Kietzmann et al., 2011*; *Geuter et al., 2014*). Pupil data were down-sampled offline from 1000 Hz to 250 Hz. Periods of ±100 ms around blinks automatically detected by the EyeLink software were removed. Additionally, we removed blinks not detected by the EyeLink software (including the intervals ± 100 ms around blinks). Trials with more than 50% of the samples missing were excluded from further analyses; 10.8% of trials had to be discarded. Missing data due to blinks were then linearly interpolated and pupil diameter traces were smoothed with a low-pass filter using a cutoff frequency of 2 Hz. Interpolating over all missing samples and analyzing all trials revealed almost identical results to the original results. The pupil results are thus independent of the eye blinks and other artifacts. Response amplitudes were computed as the difference between the maximum following stimulus onset and a 1 s pre-stimulus baseline. Amplitudes were also log-transformed before further analyses.

## fMRI data preprocessing and subject-level models

Functional imaging data were analyzed using Matlab (v8.1) and SPM8. The first five volumes of each run were discarded and the remaining images were spatially realigned for motion correction before non-linear spatial normalization using DARTEL, a high-dimensional warping algorithm available in SPM (*Ashburner, 2007*). The functional images were spatially smoothed using a Gaussian kernel with a full-width-half-maximum of 6 mm, which is three times the voxel-size.

Subject-level models included separate regressors for each of the six experimental conditions (2 temperatures × 3 cues). The predictive coding model assumes the fMRI signal to be a weighted sum of the prediction and prediction error (PE). Furthermore, PE are expected to continuously update the internal model generating the predictions (*Egner et al., 2010*; *Büchel et al., 2014*; *Summerfield and de Lange, 2014*). Predictions and PEs are thus not separable with the temporal resolution of standard fMRI and are hence modeled over the whole period from cue onset to stimulus offset (1.8 s duration) before convolution with the canonical hemodynamic response function (*Egner et al., 2010*). Additional regressors modeled the rating period and the responses to color-changes of the fixation dot. In addition, each subject-level model included six motion parameters estimated during realignment as well as the first two principal components of the time-series extracted from white matter and cerebro-spinal fluid masks as nuisance regressors.

Contrasts testing the two main effects of stimulus (painful stimulus > warm stimulus) and cue (cue high > cue low) as well as the interaction effect ((cue high, warm) > (cue low, warm)) > ((cue low, pain) > (cue high, pain)) were computed on the subject level. For group-level inference, we tested the subject level contrasts using nonparametric permutation tests (*Holmes et al., 1996*; *Nichols and Holmes, 2002*) as implemented in Statistical Non-Parametric Mapping (SnPM; http://www.warwick.ac.uk/snpm) with 6 mm variance smoothing (*Nichols and Holmes, 2002*). This tests makes less assumptions about the fMRI data than parametric analyses, while adequately controlling the whole brain family wise error rate (*Eklund et al., 2016*).

To fit the competing psychological models to the fMRI data, we used the contrast estimates for the six experimental conditions (either averaged within region of interest [ROI] or across voxels). Model details and fitting techniques are described below. Note that we did not carry out any correlational analyses between ratings and fMRI data, as the additional variance in the ratings (on whether a stimulus was perceived as warm or painful) is minimal when compared to the actual stimulation: only 5% of the responses did not match the stimuli and 18% of these mismatching responses were given on the first trial of an fMRI run.

## Neurological pain signature analyses

The Neurological Pain Signature (NPS) is a multivariate pattern of brain activity with high sensitivity and specificity in distinguishing experimental pain from other conditions like pain anticipation, pain rating periods, or vicarious pain (*Wager et al., 2013*; *Krishnan et al., 2016*). The NPS expression as a surrogate for heat pain intensity associated with a given fMRI image is computed by taking the dot-product of the NPS and the image, resulting in a scalar value. We computed NPS expression values for each of the six experimental conditions based on the regressors described above separately

for each participant (*Wager et al., 2011*, *2013*; *Krishnan et al., 2016*). The resulting NPS values were then plotted and submitted to further analyses the same way as the ROI averages (see below).

## Region of interest masks

Anatomical masks for pain processing regions of interest were generated using the Harvard-Oxford Atlas (*Desikan et al., 2006*) freely distributed with the FSL software (https://fsl.fmrib.ox.ac.uk/fsl/fslwiki/Atlases). For each hemisphere, we thresholded the probability maps for insula, parietal oper-culum (SII), post-central sulcus (SI), anterior cingulate, amygdala, and thalamus at 50%. Anterior and posterior insula masks were created by splitting the insula mask at MNI y = 0. The anterior cingulate mask of this atlas includes BA24 and BA32, but excludes subgenual portions at the set threshold of 50%. The thalamic mask covers the entire thalamus. The division of insular cortex at y = 0 mm was chosen because no probabilistic atlas includes separate maps for anterior and posterior insula. The division at y = 0 mm is close to the sulcus centralis insulae (*Nieuwenhuys, 2012*) and has been used previously (*Ploner et al., 2011*). For the periaqueductal gray (PAG), we manually created a mask based on the mean anatomical image of all subjects (*Stein et al., 2012*). For this, we first identified the central aqueduct on the mean anatomical image (shown in *Figures 4* and *5*). We then manually marked the gray matter surrounding the aqueduct and validated the resulting mask using a brain-stem atlas (*Naidich et al., 2009*).

## Models of pain processing

The first model tested here is a pure stimulus intensity-coding model in which physiological responses are a simple function of the stimulus input:

$$\hat{y} = wS \tag{1}$$

where $S$ is the stimulus intensity (dummy-coded with 0 for warm and one for pain stimuli) and $w$ is a free scaling parameter. Please note, that we do not make assumptions about the stimulus response function here. Due to the dummy-coding, the free parameter $w$ describes the mean distance between the responses to warm and painful stimuli. The distance can be determined by an arbitrary stimulus-response function, since only two stimulus intensities are used here. Expectation (*Figure 1A*; cues on the *x*-axis) has no effect on the measured response.

The second – stimulus intensity plus expectation – model (*Figure 1B*) assumes that pain responses are based on two additive effects of the expected pain plus the actual stimulation inten-sity and is described by the following formula:

$$\hat{y} = w_1 S + w_2 P \tag{2}$$

where $S$ is again the stimulus intensity, dummy-coded as in *Equation (1)* and $P$ is the expected pain as determined by the pain probability following each of the three cues (i.e., 0.25, 0.5, or 0.75). The weights $w_1$ and $w_2$ are free parameters controlling the weighting of input parameters. Parameter $w_1$ controls the distance between the two lines denoting warm and pain stimuli and accommodates any stimulus-response function in the current design with two intensities (due to the dummy coding). The expectation to receive a painful stimulus is assumed to have an additive, linear effect on the measured response. Hence, the basic relationship between stimulus intensity and response could have any form, but would be subject to linear modulation based on expectations.

Finally, the predictive coding model states that the physiological responses (fMRI parameter esti-mates or SCR or pupil dilation) are the weighted sum of the prediction (P) and the prediction error (PE; *Figure 1C*):

$$\hat{y} = w_1 P + w_2 PE \tag{3}$$

where $P$ is the expected pain (corresponding to the actual probabilities used in the experiment, that is, 0.25, 0.5, and 0.75), and $w_1$ and $w_2$ are free parameters. $PE$ is the difference between pain outcome and prediction (i.e., 1-$P$), if the outcome is painful. In the case of non-painful warmth, the $PE$ is 0. We chose this $PE$ formulation based on (1) previous studies in the visual system (*den Ouden et al., 2010*; *Egner et al., 2010*), (2) the assumption that pain specific populations will only encode prediction errors for painful stimuli (*Belova et al., 2007*; *Büchel et al., 2014*), and (3) the

observation that prediction errors for painful and non-painful warmth have different topographies (*Ploghaus et al., 2000*; *Zeidan et al., 2015*).

In addition to this PE definition, we considered a signed *PE* in which $PE = S - P$ and an absolute *PE* model in which $PE = |S - P|$ (i.e., high prediction error for both, unexpected pain and unexpected non-painful stimuli). Again, *S* being the dummy-coded stimulus intensity and *P* being the expected probability of receiving pain as above. Those *PE* definitions are based on observations in the visual (*Kok et al., 2014*) and auditory (*Todorovic et al., 2011*; *Todorovic and de Lange, 2012*) systems.

## Model comparison

We used Bayes Factors (BF) for pairwise comparisons testing which model offers the best explanation of the data (*Jeffreys, 1961*; *Kass and Raftery, 1995*; *Lee and Wagenmakers, 2014*). Bayes Factors formulate evidence for one model over the other as the ratio of the two marginal likelihoods; that is the likelihood of the data under each of the models integrated over the model's parameter space, respectively. A Bayes Factor can be interpreted as 'how much more likely is model A compared to model B?' For example, a Bayes Factor of four indicates that model A is four times as likely as B to have generated the data, whereas a Bayes Factor of 0.1 indicates that B is ten times as likely as A. Furthermore, Bayes Factors select the most predictive model and implicitly penalize model complexity. Another benefit of using Bayes Factors is their ability to compare non-nested models (*Lee and Wagenmakers, 2014*). We used the BayesFactor package (v. 0.9.11) for R by Rouder and Morrey (*Rouder and Morey, 2012*) to compute Bayes Factors (two chains, each with 80,000 samples with thinning factor of four for each estimation). This implementation uses default mixture-of-variance (Cauchy) priors on the weight parameters that have desirable properties of the resulting Bayes factors (location and scale invariance, consistency, and consistent in information) (*Rouder and Morey, 2012*). We use log-BF throughout the manuscript to ease the interpretation, because log-BF favoring one or the other model have different signs, but the same scaling: A log-BF of four indicates the same amount of evidence for model A, as a log-BF of $-4$ does for model B. A value of $|logBF| > 3$ indicates that one model is ~20 times more likely than the alternative model and is conventionally labeled as 'strong support' for a given model (*Kass and Raftery, 1995*; *Stephan et al., 2010*; *Lodewyckx et al., 2011*). Multiple model comparisons using Bayes factors do not need an explicit correction as is necessary in frequentist approaches (*Scott and Berger, 2006*).

Bayes Factors were computed for the log-transformed amplitudes of SCR and pupil responses, as well as for the average parameter estimates extracted from our anatomical ROIs. To achieve better spatial resolution, we also computed Bayes Factors for brain voxels within a mask defined by an omnibus F-test for a non-zero effect of any condition (thresholded at p<0.005, uncorrected). Following the above introduced convention for strong model support by log-BF, we display voxels with $|logBF| > 3$.

Log-BF maps were overlaid on the group-mean anatomical image using Matlab functions from Tor D. Wager's group (https://github.com/canlab). We used Caret 5 (v. 5.65, http://brainmap.wustl.edu/caret.html) for surface visualization of the log-BF insula map.

## Acknowledgements

We thank Tor D Wager for sharing the NPS pattern. SG was supported by the DFG (GE 2774/1–1). CB was supported by the DFG (SFB 936, project A06), and the ERC (2010-AdG_20100407). The authors declare no conflicts of interest.

## Additional information

### Funding

| Funder | Grant reference number | Author |
|---|---|---|
| Deutsche Forschungsgemeinschaft | SFB 936 A06 | Christian Büchel |
| European Commission | ERC Advanced Investigator Grant 2010-AdG_20100407 | Christian Büchel |

| Deutsche Forschungsge-meinschaft | Fellowship GE 2774/1-1 | Stephan Geuter |
|---|---|---|

The funders had no role in study design, data collection and interpretation, or the decision to submit the work for publication.

### Author contributions

SG, Conceptualization, Data curation, Formal analysis, Investigation, Visualization, Methodology, Writing—original draft; SB, Conceptualization, Investigation, Methodology, Writing—review and editing; FE, Conceptualization, Methodology, Writing—review and editing; CB, Conceptualization, Resources, Supervision, Funding acquisition, Methodology, Writing—review and editing

### Author ORCIDs

Stephan Geuter, http://orcid.org/0000-0002-4935-5692
Falk Eippert, http://orcid.org/0000-0002-3986-1719

### Ethics

Human subjects: The study was approved by and conducted in accordance with the ethics guidelines of the Medical Chamber Hamburg (PV4745). All participants provided informed consent to participate and to publish.

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
