## [Decision Letter]

[Editors’ note: a previous version of this study was rejected after peer review, but the authors submitted for reconsideration. The first decision letter after peer review is shown below.]

Thank you for submitting your work entitled "From intensity-coding to predictive coding in pain perception" for consideration by *eLife*. Your article has been reviewed by two peer reviewers, and the evaluation has been overseen by Heidi Johansen-Berg as the Reviewing Editor and Jody Culham as the Senior Editor. Our decision has been reached after consultation between the reviewers. Based on these discussions and the individual reviews below, we regret to inform you that your work will not be considered further for publication in *eLife*.

*Reviewer #1:*

This is an interesting paper that seeks to understand how predictive coding impacts the link between brain responses and perceptual responses to painful heat stimuli. The authors describe a predictive coding model that represents the weighted sum of a predicted stimulus plus a prediction error (that can be manipulated by different cues). I found this to be a most interesting topic that certainly provides a framework from which to think about pain experiences and chronic pain perhaps. However, I did find the paper quite difficult to understand without a background in this type of modeling and Bayesian analyses and also because there were many methodological details missing. Furthermore, I think that their data acquisition/analysis, especially of the brainstem, may not be adequate.

1) The Introduction needs to describe what a Bayes approach is for non-experts.

2) The figures need some work in their presentation and also the concepts described in the legends:

Figure 1 and the concept shown makes a big assumption about pain processing regions increasing activity with intensity. But this may not be the case and the relationship may not be linear at all. The stimulus-response curves for different types of nociceptive neurons is not necessarily linear and so the premise of the model may be flawed. The authors need to explain their assumptions better.

Figure 2 needs some modifications so the reader can understand the magnitude of the effects; so clearly indicate the units on all axes and explain why a log scale is used. It is also important to show/state what the SCR and pupil size are without/before the stimulus/cues.

Figure 2—figure supplement 1 should show the error bars to reveal the intersubject variability and also change the color of the lines as they are not distinguishable. Also, the data should also be shown with the alignment at the stimulus onset.

3) The calibration methods are not fully described and so it is not possible to fully understand the protocol. For example:

What was the temperature used (only a range of 42-49.5 was noted)?

What order were they delivered (ascending, random?)

What was the rise time?

What was the interstimulus interval?

How many repetitions of each temperature were there?

The VAS is non-standard and not clearly described. How were the subjects instructed to use this (e.g., for the half of the scale between "no sensation" and pain threshold"?

4) Was there a sex difference in the thresholds determined in the calibration phase?

5) The Methods (subsection “Procedure”, end of second paragraph) states that the average temp for the pain stimulus was 49.4 +/-1.3C (SD). This range is impossible since the maximal stimulus was 49.5C. Please check this since it must be an error.

6) Where was skin conductance measured?

7) What is the purpose of a delay and it being a variable interval of 3-5s after the stimulus and before the cue to rate?

8) There is a design problem in ratings of perceived intensity are only obtained after the entire fMRI block. Therefore, there is no way to know what the subjects are really evaluating – their memory of pain to last the stimulus? Average pain? Also, since the block had both warm and pain trials, these would likely have varied across 48 trials. The authors could have run a psychophysical session first to just get pain/warm ratings to each of the 48 trials to see how consistent they were (likely quite variable from beginning to end of the session).

9) The authors should justify their choice of using an MRI acquisition with a gap which could be problematic (especially in the brainstem).

10) More than 10% of trials were discarded due to blinks. This sounds like a lot of missing data. Please comment.

11) The method for obtaining and analyzing pupil data is likely not known by most readers and so there could be more detail and previous papers should be cited to validate whether this is a standard method.

12) I am not sure how the fMRI data were analyzed and I have reservations whether adequate methodology was used to detect findings in the brainstem (PAG): Did you use linear or non-linear registration? Did you use motion correction? Did you use a smaller smoothing kernel for the brainstem?

13) I think it would be important to analyze the responses to cue and stimulus separately rather than (or in addition to) "modelining the whole period from cue onset to stimulus offset".

14) The brain location of the findings need to be better stated and labelled so that the reader understands the exact part or subregion of the area referred to, such as in the insula, anterior cingulate, and thalamus. Also, the authors state they divided the insula into 2 parts at y=0 but do not give a reason for this approach. Also, the method to "manually creating a mask" for the PAG should be described.

*Reviewer #2:*

The study by Geuter et al. entitled "From intensity-coding to predictive coding in pain perception" attempts to address a very interesting and important question: which brain mechanisms transform nociceptive input into an experience of pain, and how does this occur within a Bayesian framework? Significant methodological issues raise substantial concerns about the conclusions that can be drawn from this study.

1) Why was the cue delivered only 300ms prior to the stimulus? Is there any evidence that a cue presented so shortly before the stimulus could be consciously perceived and interpreted? More specifically, does this cueing paradigm allow subjects to develop an adequate prediction of the impending stimulus?

2) More importantly, why was the cue not terminated at stimulus onset? When the cue and the stimulus are co-administered, the task changes dramatically from one of prediction to one in which concurrent discordance of cue and stimulus is being evaluated.

3) Similarly, the modeling of the cue onset to stimulus offset is problematic in that it does not allow the separation of cue related activity from stimulus related activity.

4) Warm sensitive primary afferents are extremely slowly conducting C fibers (0.5-2m/s). The stimulus-related regressor does not appear to be adjusted to take the conduction delay into account. Thus, warm stimulus-related activation is unlikely to be captured by the regressor.

[Editors’ note: what now follows is the decision letter after the authors submitted for further consideration.]

Thank you for submitting your article "Functional Dissociation of Stimulus Intensity Encoding and Predictive Coding of Pain in the Insula" for consideration by *eLife*. Your article has been reviewed by two peer reviewers, and the evaluation has been overseen by a Reviewing Editor and Sabine Kastner as the Senior Editor. The following individual involved in review of your submission has agreed to reveal his identity: Floris de Lange (Reviewer #2).

The reviewers have discussed the reviews with one another and the Reviewing Editor has drafted this decision to help you prepare a revised submission.

Summary:

The reviewers and editor recognised that the paper addressed an interesting question and approached question in an innovative way. Although some improvements were noted from the previous version of this manuscript that was considered, there were still some key issues that required addressing.

After consultation between reviewers, we propose the following essential revisions:

Essential revisions:

1) Linearity of response

Figure 1: The authors maintain that their mode only assumes that the physiological response is stronger for more intense stimulation. I disagree. As I noted in my previous review, there are nociceptive neurons that do not necessary have linear increases to increasing stimulus intensity. The way Figure 1 is shown, it indeed depicts a linear response. Furthermore, there are neurons that do not have increasing responses to increasing stimulus intensity – they can plateau and max out at particular levels of stimulation. This needs clarification.

2) There are a number of key, relevant concepts that are not discussed. The authors should at least cite and compare their finding to these important papers:

A) noxious intensity coding in the cortex and insula functionality and connectivity

Baliki MN, Geha PY, Apkarian AV. Parsing pain perception between nociceptive representation and magnitude estimation. J Neurophysiol. 2009;101:875-887.

Taylor KS, Seminowicz DA, Davis KD. Two systems of resting state connectivity between the insula and cingulate cortex. Hum Brain Mapp. 2009;30:2731-2745

Seminowicz DA1, Davis KD. Interactions of pain intensity and cognitive load: the brain stays on task. Cereb Cortex. 2007 Jun;17(6):1412-22.

Coghill RC, Sang CN, Maisog JM, Iadarola MJ. Pain intensity processing within the human brain: a bilateral, distributed mechanism. J Neurophysiol. 1999 Oct;82(4):1934-43

B) The concept of salience is also not discussed and this is important because responses to increasing noxious stimulus intensity could simply reflect salience responses. The authors thus need to discuss their findings and the salience issue in light of the many findings from the Iannetti lab and the Downar papers from the Davis lab.

C) The concept of comparing stimulus-encoding with perceptual-encoding is not new and the authors should acknowledge the foundational work done by several labs that addressed this issue of percept-related fMRI over the last 15 years – namely the Apkarian lab, the Davis lab, and the Porro lab. The key papers are reviewed in Davis KD, Moayedi M. Central mechanisms of pain revealed through functional and structural MRI. J Neuroimmune Pharmacol. 2013 Jun;8(3):518-34

3) Modeling of PEs:

The authors write: "In the case of non-painful warmth, the PE is 0."

How valid is this choice? In their model, if there is a strong prediction of painful stimulus, followed by an absence of a painful stimulus, the PE is zero. This stands in contrast to observations in the reward system – where a strong reward expectation, followed by an absence of a reward leads to a negative RPE, and a corresponding dip in activity, i.e. signed RPE. It also stands in contrast to observations in the visual system – where a strong stimulus expectation, followed by an absence of the stimulus leads to an enhanced visual response (den Ouden et al., 2008; den Ouden et al., 2010; Kok et al. 2014). And it stands in contrast to observations in the auditory system – where an expected but omitted tone leads to stronger activity (Todorovic & de Lange 2012). I understand that the author's model provides a better fit to the data. Would it be possible to include a model into the comparison in which PEs are also generated for 'pain absent' trials, so that these can also be formally compared? At least the authors should discuss this issue, as it can by no means be taken for granted that no PEs are generated for 'stimulus absent' trials.

4) Subjective perception of pain:

Looking at the results, it is true that the SCR, pupil dilation and BOLD signal can be modeled using a predictive coding model similar to the one used by Egner et al. (2010). However, along the whole study, these results are not linked in any manner to the participant's subjective perception of pain. Previous studies investigating predictive coding showed that predicting sensory information enhances the representation of the predicted input and therefore its perception. When reading the manuscript, some immediate questions come to mind: Does predictive pain enhance perception of pain compared to unpredicted pain? What would an enhanced representation of pain imply? (more or less subjective pain?) When people successfully predict a painful stimulus, is the stimulus perceived more or less painful?

5) Given the two alternatives design (painful Y/N) and that warm/painful stimuli were clearly dissociable, there may not be enough variance in the data in order to correlate subjective pain ratings with the changes in brain activity across experimental conditions. Showing the behavioral correlates of each condition may nevertheless improve the significance of the study.

---

## [Author Response]

[Editors’ note: the author responses to the first round of peer review follow.]

*Reviewer #1:*

*This is an interesting paper that seeks to understand how predictive coding impacts the link between brain responses and perceptual responses to painful heat stimuli. The authors describe a predictive coding model that represents the weighted sum of a predicted stimulus plus a prediction error (that can be manipulated by different cues). I found this to be a most interesting topic that certainly provides a framework from which to think about pain experiences and chronic pain perhaps. However, I did find the paper quite difficult to understand without a background in this type of modeling and Bayesian analyses and also because there were many methodological details missing. Furthermore, I think that their data acquisition/analysis, especially of the brainstem, may not be adequate.*

In order to stay within the limits of the Short Reports format, we had to restrict the explanation of some of the ideas and methods. Encouraged by the reviewer’s feedback and suggestions, we have expanded the manuscript into a full-length article and now devote more space to the introduction of the concepts and details of the analyses.

In addition, we extended our analyses and now include a third alternative model based on important research on expectation-based pain modulation. This model explains physiological responses through additive effects of stimulation intensity and expectation (stimulus plus expectation model).

Previous studies from our laboratory using the same fMRI sequence and similar analyses have demonstrated the sensitivity of those procedures for brainstem imaging nuclei (Eippert et al., 2009; Aderjan et al., 2010; Stankewitz et al., 2011; Sprenger et al., 2015). The present acquisition and analyses techniques are thus well suited for brainstem analyses.

*1) The Introduction needs to describe what a Bayes approach is for non-experts.*

Reformatting the manuscript into a regular length article provided us with the space to describe the Bayesian approach and methods in more detail (see Introduction and Results sections).

*2) The figures need some work in their presentation and also the concepts described in the legends:*

We have substantially re-organized the results presented in the figures to enhance clarity and included the reviewer’s suggestions mentioned below.

*Figure 1 and the concept shown makes a big assumption about pain processing regions increasing activity with intensity. But this may not be the case and the relationship may not be linear at all. The stimulus-response curves for different types of nociceptive neurons is not necessarily linear and so the premise of the model may be flawed. The authors need to explain their assumptions better.*

We agree with this reviewer that the stimulus-response curves for different types of nociceptive neurons is not necessarily linear, however, this does not affect our models, because the only assumptions we make are i) that the physiological response is stronger for the more intense stimulation, and ii) that the cue has no effect on the brain response under this model (flat lines in Figure 1). The difference between the two stimuli is a free parameter that is estimated from the data, i.e. the average distance between responses for warm and painful stimuli is allowed to vary according to the free parameter.

For the predictive coding model, we assume that the response scales with the expected probability for pain, a reasonable assumption based on multiple studies reporting linear scaling of expected painful stimuli (Seymour et al., 2004, 2005; Atlas et al., 2010). We explain these assumptions now in more detail in the Methods section.

*Figure 2 needs some modifications so the reader can understand the magnitude of the effects; so clearly indicate the units on all axes and explain why a log scale is used. It is also important to show/state what the SCR and pupil size are without/before the stimulus/cues.*

We computed evoked responses according to common standards and guidelines (Alvarez et al., 2008; Feld et al., 2010; Boucsein et al., 2012; Boll et al., 2013; Geuter et al., 2014). All measures of interest are relative to a fluctuation pre-cue baseline and are hence corrected for this baseline. Most commercially available eye-tracker, including our model, record the pupil size in arbitrary units (pixels). Because the transformation to mm^2^ is linear, we did not include a pupil-size calibration procedure as we have done before outside the MR scanner (Geuter et al. 2014). The units for pupil dilation are thus arbitrary. SCR units are measured in µSiemens, a measure of electric conductivity: G=IV.

Logarithmic transformation increase the normality of empirical SCR and pupil diameter distributions and are standard for experiments investigating evoked responses (Society for Psychophysiological Research Recommendations in Boucsein et al. 2012).

*Figure 2—figure supplement 1 should show the error bars to reveal the intersubject variability and also change the color of the lines as they are not distinguishable. Also, the data should also be shown with the alignment at the stimulus onset.*

These panels are now shown in Figure 2. We increased the luminance differences between the colors for better readability and now show the standard error of the mean as shaded areas. Because of the fixed interval between cue and heat stimulus onset, we set cue onset as t=0. Heat stimulus onset is at t=0.3s and we now show a tick mark at this time-point.

*3) The calibration methods are not fully described and so it is not possible to fully understand the protocol. For example:*

We extended the Methods section and now include all requested information. Please find the information also copied below.

*What was the temperature used (only a range of 42-49.5 was noted)?*

Temperatures between 42-49.5°C in steps of 0.5°C resulting in a total of 16 different temperatures.

*What order were they delivered (ascending, random?)*

The order was pseudo-randomized.

*What was the rise time?*

The temperature ramp was 70°C/s, resulting in ramp-up times of 140 – 250msec.

*What was the interstimulus interval?*

The inter-stimulus interval was 13-17 sec plus the time participants used for the rating (mean: 5.04 sec).

*How many repetitions of each temperature were there?*

Each temperature was repeated two times, except for 44, 45, 46, and 47°C which were repeated three times.

*The VAS is non-standard and not clearly described. How were the subjects instructed to use this (e.g., for the half of the scale between "no sensation" and pain threshold"?*

Participants were instructed to rate all heat stimuli that did not feel painful below the ‘pain threshold’ mark. In case they did not feel any change from baseline temperature (32°C), they were instructed to rate this as ‘no sensation at all’. The average correlation between temperature and rating was high with a mean r = 0.78 (std: 0.13). We expanded the description of our scale in the Methods section.

*4) Was there a sex difference in the thresholds determined in the calibration phase?*

There was no significant sex difference in pain sensitivity (t(26) = 1.32; p = 0.2). We now report this result in the manuscript.

*5) The Methods (subsection “Procedure”, end of second paragraph) states that the average temp for the pain stimulus was 49.4 +/-1.3C (SD). This range is impossible since the maximal stimulus was 49.5C. Please check this since it must be an error.*

Indeed, the maximal temperature was capped at 49.5°C. The temperature distribution is thus cutoff at this level and non-normally distributed. We reported the standard deviation to characterize the variability in temperatures used. In order to clarify that we do not imply temperatures above 49.5°C, we now phrase the sentence as follows:

“The average temperature for the warm stimulus was 45.0°C (standard deviation: 1.2°C) and the average temperature for the painful stimulus was 49.4°C (standard deviation: 1.3°C) with a maximum temperature of 49.5°C.”

*6) Where was skin conductance measured?*

Skin conductance was recorded from the thenar and hypothenar eminences of the left hand. We now report this in the Methods section.

*7) What is the purpose of a delay and it being a variable interval of 3-5s after the stimulus and before the cue to rate?*

A variable jitter between two events is necessary in order to reliably estimate BOLD responses to each event (Dale, 1999; Friston et al., 1999); we have added this information to the manuscript.

*8) There is a design problem in ratings of perceived intensity are only obtained after the entire fMRI block. Therefore, there is no way to know what the subjects are really evaluating – their memory of pain to last the stimulus? Average pain? Also, since the block had both warm and pain trials, these would likely have varied across 48 trials. The authors could have run a psychophysical session first to just get pain/warm ratings to each of the 48 trials to see how consistent they were (likely quite variable from beginning to end of the session).*

We agree that the delay between stimulation and VAS is not ideal. However, in order to minimize the time for participants spend in the scanner we did not present a VAS after each trial. Instead, we asked after each trial whether the stimulus was perceived as painful or not (‘yes’ or ‘no’). In 94.3% of the trials, the percept matched the stimulus. Excluding the non-matching trials led to very similar response patterns in each brain region.

Furthermore, the VAS ratings were stable across blocks. We now include more information on these validity checks in the manuscript.

Author response image 1.Intensity ratings after each fMRI block.**DOI:**
http://dx.doi.org/10.7554/eLife.24770.011

*9) The authors should justify their choice of using an MRI acquisition with a gap which could be problematic (especially in the brainstem).*

The use of a small inter-slice gap is common in fMRI experiments to prevent side-excitations from neighboring slices. This sequence is well established and our laboratory has shown that it is sensitive to responses in small brainstem nuclei (Eippert et al., 2009; Aderjan et al., 2010; Stankewitz et al., 2011; Sprenger et al., 2015).

*10) More than 10% of trials were discarded due to blinks. This sounds like a lot of missing data. Please comment.*

According to our experience, this amount of missing trials is within the expected range for an hour-long pain study when all blinks are being removed from the data (Geuter et al., 2014). Unfortunately, most studies measuring pupil dilation do not report the amount of discarded trials and it is thus difficult to compare this amount of missing data to other studies.

Interpolating over all missing samples and plotting pupil responses for all trials revealed a very similar result to the original results (see Figure 9). The pupil results are thus independent of the eye blinks and other artifacts.

Author response image 2.Pupil responses based on all trials (left panel) and after exclusion of trials with artifacts (right panel).**DOI:**
http://dx.doi.org/10.7554/eLife.24770.012

*11) The method for obtaining and analyzing pupil data is likely not known by most readers and so there could be more detail and previous papers should be cited to validate whether this is a standard method.*

Pupillometry analyses followed standard procedures (Einhäuser et al., 2008; Hupé et al., 2009; Kietzmann et al., 2011; Geuter et al., 2014) and we clarified this in the manuscript.

*12) I am not sure how the fMRI data were analyzed and I have reservations whether adequate methodology was used to detect findings in the brainstem (PAG): Did you use linear or non-linear registration? Did you use motion correction? Did you use a smaller smoothing kernel for the brainstem?*

We used a high-dimensional, nonlinear spatial normalization algorithm (DARTEL; Ashburner, 2007) and functional images were motion corrected. Due to its high- dimensional optimization algorithm, DARTEL provides accurate brainstem registration. We also explored a brainstem-only preprocessing (including a smaller smoothing kernel) and analysis as previously described (Napadow et al., 2006; Eippert et al., 2009), but did not observe additional activations. We clarified the respective paragraph in the Methods section.

*13) I think it would be important to analyze the responses to cue and stimulus separately rather than (or in addition to) "modelining the whole period from cue onset to stimulus offset".*

We chose to present the cues and stimuli in close temporal succession because the predictions and prediction errors are expected to continuously update predictions while being passed along the processing hierarchy (Egner et al., 2010; Büchel et al., 2014; Summerfield and de Lange, 2014). They are thus not separable with the temporal resolution of standard fMRI and we thus modeled the whole period from cue onset to stimulus offset (1.8 s duration) as block regressors before convolution with the canonical hemodynamic response function following Egner et al. (2010).

*14) The brain location of the findings need to be better stated and labelled so that the reader understands the exact part or subregion of the area referred to, such as in the insula, anterior cingulate, and thalamus. Also, the authors state they divided the insula into 2 parts at y=0 but do not give a reason for this approach. Also, the method to "manually creating a mask" for the PAG should be described.*

We defined our regions of interest based on the Harvard-Oxford Atlas (https://fsl.fmrib.ox.ac.uk/fsl/fslwiki/Atlases). The anterior cingulate mask of this atlas includes BA24 and BA32, but excluding subgenual portions at the set threshold of 50%. The thalamic mask covers the entire thalamus. The division of insular cortex at y=0mm was chosen because this atlas does not include separate maps for anterior and posterior insula. The division at y = 0 mm is at the sulcus centralis insulae and has been previously used (Ploner et al., 2011). We did not reprint the masks to save space and atlas maps are freely available online. Furthermore, their development has been described in previous publications (Desikan et al., 2006; Goldstein et al., 2007), and 2) we also show voxel-wise results for all analyses that allow a more precise localization of effects. However, we would be happy to provide images showing the extent of all our masks.

For the PAG mask, we used an approach similar to the one used by Stein et al. (2012). We first identified the central aqueduct on the mean anatomical image. We then manually marked the gray matter surrounding the aqueduct and validated the resulting mask using a brainstem atlas (Naidich et al., 2009). Please see the figure below for the PAG mask.

We now describe the masks and the creation of the PAG mask in more detail in the manuscript.

Author response image 3.PAG mask overlaid on the group-mean structural image.**DOI:**
http://dx.doi.org/10.7554/eLife.24770.013

*Reviewer #2:*

*The study by Geuter et al. entitled "From intensity-coding to predictive coding in pain perception" attempts to address a very interesting and important question: which brain mechanisms transform nociceptive input into an experience of pain, and how does this occur within a Bayesian framework? Significant methodological issues raise substantial concerns about the conclusions that can be drawn from this study.*

*1) Why was the cue delivered only 300ms prior to the stimulus? Is there any evidence that a cue presented so shortly before the stimulus could be consciously perceived and interpreted? More specifically, does this cueing paradigm allow subjects to develop an adequate prediction of the impending stimulus?*

The present design closely follows a previous study on predictive coding in visual perception (Egner et al., 2010). That study used a cue presentation time of 250ms and demonstrated significant effects of this cue with regards to predictions and prediction errors. Furthermore, there is abundant evidence that a window of 300ms is sufficient to process and categorize the visual cue: i) psychophysical studies have shown that humans can correctly categorize visual scenes presented for only 53ms even without paying attention to these scenes (Li et al., 2002), ii) humans can also detect animals in pictures presented for 20ms with median response times below 300ms (VanRullen and Thorpe, 2001), and iii) scalp EEG potentials differentiate between categories of visual stimuli as early as 80ms after stimulus onset and frontal potentials reflect the response decision around 150ms after stimulus onset (Thorpe et al., 1996; VanRullen and Thorpe, 2001).

In any case, please note that predictive coding models do not require the internal predictions to be conscious (Knill and Pouget, 2004; Friston, 2005). We have extended the discussion of this question in the revised manuscript.

*2) More importantly, why was the cue not terminated at stimulus onset? When the cue and the stimulus are co-administered, the task changes dramatically from one of prediction to one in which concurrent discordance of cue and stimulus is being evaluated.*

It is standard practice when investigating predictions and predictions errors that the cue is still present during the outcome – this is not only the case when investigating perceptual prediction errors as done here, but also when investigating reward prediction errors (such as Wolfram Schultz’s and others’ work on dopamine neurons) or in fear learning studies that almost always present the shock together with a cue (e.g. Büchel et al., 1998; Phelps et al., 2004; Haaker et al., 2013). Below, we elaborate on this issue in detail.

Predictive coding models propose that predictions generated from an internal model are compared against incoming sensory signals. The internal model will incorporate available information in its predictions and revise predictions based on new information or emerging prediction errors. The removal of the cue would provide no additional information and would thus not change predictions or expected physiological activity (Friston, 2005; Büchel et al., 2014; Summerfield and de Lange, 2014).

Another consequence of this model is that a strict temporal dissociation between prediction and prediction error is not possible, because predictions and prediction errors are computed continuously. However, each of the two components will only change once new information (cue or stimulus) arrives. We thus followed Egner et al. (2010) and deliberately modeled cue and stimulus together.

*3) Similarly, the modeling of the cue onset to stimulus offset is problematic in that it does not allow the separation of cue related activity from stimulus related activity.*

The reviewer is correct in stating that we cannot separate cue-related from stimulus-related activity. However, such a separation is not necessary for our research question since all models that we compared make clear predictions what type of physiological responses are to be expected for the combination of cue and stimulus. We thus decided to present the cue and stimulus in brief temporal succession in order to test the hypothesis that the BOLD signal reflects the sum of prediction and prediction error.

By using a short and constant delay between the two events and prior training, we ensured that the brain would be able to predict both the most likely temperature and its onset. Based on the fact that the BOLD signal acts as a temporal low-pass filter on neuronal activity, we can test additive and interactive effects of cue-related and stimulus-, or prediction error-related activity. A longer and jittered delay between cue and stimulus would reduce the precision of the prediction and hence the resulting prediction error. The present design thus maximizes the experimental sensitivity and is based on theoretical considerations discussed in comment 2) above.

We now explain the rationale for this design in more detail in the revised Results section.

*4) Warm sensitive primary afferents are extremely slowly conducting C fibers (0.5-2m/s). The stimulus-related regressor does not appear to be adjusted to take the conduction delay into account. Thus, warm stimulus-related activation is unlikely to be captured by the regressor.*

This is an important point. However, the BOLD signal is also relatively slow. We computed a peri-stimulus time histogram for the warm and painful stimuli from an 8mm sphere located in the posterior insula and parietal opercular region (see Figure 11). There was no difference between the peak timing for both conditions.

Author response image 4.Evoked BOLD response in posterior insula for warm and painful stimuli.Responses from a sphere centered at MNI coordinates [40/-16/16] are aligned to cue onset at t=0.**DOI:**
http://dx.doi.org/10.7554/eLife.24770.014

[Editors' note: the author responses to the re-review follow.]

*Essential revisions:*

*1) Linearity of response*

*Figure 1: The authors maintain that their mode only assumes that the physiological response is stronger for more intense stimulation. I disagree. As I noted in my previous review, there are nociceptive neurons that do not necessary have linear increases to increasing stimulus intensity. The way Figure 1 is shown, it indeed depicts a linear response. Furthermore, there are neurons that do not have increasing responses to increasing stimulus intensity – they can plateau and max out at particular levels of stimulation. This needs clarification.*

We completely agree that not all types of nociceptive neurons increase their firing rate linearly with increasing stimulus intensity. Within the present paradigm, we cannot and do not try to differentiate between linear or other types of stimulus-response functions. In order to compare linear to non-linear response functions, we would need at least 3 stimulus levels, but we only have two stimulus levels (non-painful warmth and painful heat). Figure 1 does not show a linear response for stimulation – it only shows a difference between warm and painful stimulation. The linearity the reviewer refers to is only related to the prediction / expectation component that is conferred by the cue (which has three levels).

We apologize that this was not clear enough in the previous version of our manuscript. We hope to have remedied this by clarifying this in the Methods section of our manuscript (Models of pain processing). Figure 1 shows the predicted BOLD response for the stimulus-only model. Here, the expectation (cues on the x-axis) has no effect on the BOLD response, resulting in flat lines for warm and painful heat stimuli. The distance between the warm and heat stimuli is free to vary and the distance between the two lines is estimated from the data. Here, we use a simple regression with dummy-coding for the stimulus (warm = 0, pain = 1). The dummy-coding reduces this relationship to a test for any difference between the two conditions (like a t-test). This model thus accommodates any stimulus determined response function within our design with 2 stimulus intensities.

The same dummy-coding is used in the stimulus plus expectation model (Figure 1) – the distance between the two lines denoting warm and pain stimuli is free to vary. However, we assume that the expectation to receive a painful stimulus has an additive, linear effect on the BOLD response in a given voxel. Hence, the basic relationship between stimulus intensity and response could have any form, but would be subject to linear modulation based on expectations.

The predictive coding model (Figure 1) again assumes no direct linear relationship between stimulus intensity and outcome. Expectation is expected to linearly contribute to the BOLD signal as is the error term for the painful stimulus depending on the expectation (Seymour et al., 2005, 2007; Atlas et al., 2010; den Ouden et al., 2010; Egner et al., 2010; Zhang et al., 2016).

*2) There are a number of key, relevant concepts that are not discussed. The authors should at least cite and compare their finding to these important papers:*

*A) noxious intensity coding in the cortex and insula functionality and connectivity*

Baliki MN, Geha PY, Apkarian AV. Parsing pain perception between nociceptive representation and magnitude estimation. J Neurophysiol. 2009;101:875-887.

*Taylor KS, Seminowicz DA, Davis KD. Two systems of resting state connectivity between the insula and cingulate cortex. Hum Brain Mapp. 2009;30:2731-2745*

Seminowicz DA1, Davis KD. Interactions of pain intensity and cognitive load: the brain stays on task. Cereb Cortex. 2007 Jun;17(6):1412-22.

*Coghill RC, Sang CN, Maisog JM, Iadarola MJ. Pain intensity processing within the human brain: a bilateral, distributed mechanism. J Neurophysiol. 1999 Oct;82(4):1934-43*

We included the references above and now explicitly discuss our findings with respect to these studies.

*B) The concept of salience is also not discussed and this is important because responses to increasing noxious stimulus intensity could simply reflect salience responses. The authors thus need to discuss their findings and the salience issue in light of the many findings from the Iannetti lab and the Downar papers from the Davis lab.*

We agree that more intense stimuli will be in general more salient than less intense stimuli. The elegant studies by Downar & Davis (Downar et al., 2000, 2002, 2003) identified multiple brain regions responding to salient events, with the temporal parietal junction (TPJ) responding most consistently. Anterior insula and anterior cingulate cortex (ACC) also responded to unexpected or novel stimuli, but anterior insula did not display the expected salience response profile in the context of pain (Downar et al., 2002). Interestingly, the anterior insula in our study also shows a strong surprise or prediction error response. Another study interpreted the activation of the insula during non-painful stimulation as a supra-modal saliency response (Mouraux et al., 2011). According to Mouraux et al. (2011), saliency is determined by how much a stimulus differs from the preceding stimuli (p. 2247). This is definition is closely related to the prediction error concept in predictive coding theories.

Here, we tested the effects of predictions and prediction errors on autonomic and brain responses by manipulating cue-validity. The predictive coding model de-composes the response into two distinct functional components – a prediction term and prediction error term. From a psychological perspective, this is meaningful because people would want to direct their attention towards expected aversive events (high probability of pain, prediction term) and towards unexpected events (prediction error). Both functions are important, but the former is future-oriented, while the latter is based on past experience. The prediction error is assumed to drive learning and hence critical for adaption to future events.

An unexpected or more aversive than expected stimulus would thus stand out from previous sensory input and this could be referred to as salience. However, for this salience to be computed, a stimulus must have been processed and its nature and surprise (or prediction error) needs to be available. Salience in this context can thus be considered a secondary property of a processed stimulus that will affect future learning. This is for example formalized in the Pearce-Hall learning model in which a surprising, salient outcome, affects the learning rate in the next trial (Pearce and Hall, 1980).

We now discuss the relationship between stimulus intensity, prediction errors, and salience in the revised manuscript.

*C) The concept of comparing stimulus-encoding with perceptual-encoding is not new and the authors should acknowledge the foundational work done by several labs that addressed this issue of percept-related fMRI over the last 15 years – namely the Apkarian lab, the Davis lab, and the Porro lab. The key papers are reviewed in Davis KD, Moayedi M. Central mechanisms of pain revealed through functional and structural MRI. J Neuroimmune Pharmacol. 2013 Jun;8(3):518-34*

We extended the Introduction and Discussion to explicitly include these papers comparing stimulus and pain report correlations. This is an important literature offering insights into which brain regions are more closely related to nociceptive input and which are more closely related to subjective experiences and reports.

In this study, we decomposed the response pattern into predictions and prediction errors (PE), allowing us to investigate which computational principles link the two representational levels, i.e., stimulus and expected percept. This approach is complementary to the percept correlation approach, in that it investigates functional components of pain processing. According to predictive coding accounts, the response pattern in a brain region will reflect stimulus properties and prior expectations depending on the state of the internal model and the surprise of the sensory input. The present study thus significantly extends the seminal findings mentioned by the reviewer (Davis et al., 1998; Porro et al., 1998; Apkarian et al., 2001; Bornhövd et al., 2002; Büchel et al., 2002; Davis et al., 2002, 2004; Porro et al., 2004).

*3) Modeling of PEs:*

*The authors write: "In the case of non-painful warmth, the PE is 0."*

*How valid is this choice? In their model, if there is a strong prediction of painful stimulus, followed by an absence of a painful stimulus, the PE is zero. This stands in contrast to observations in the reward system – where a strong reward expectation, followed by an absence of a reward leads to a negative RPE, and a corresponding dip in activity, i.e. signed RPE. It also stands in contrast to observations in the visual system – where a strong stimulus expectation, followed by an absence of the stimulus leads to an enhanced visual response (den Ouden et al., 2008; den Ouden et al., 2010; Kok et al. 2014). And it stands in contrast to observations in the auditory system – where an expected but omitted tone leads to stronger activity (Todorovic & de Lange 2012). I understand that the author's model provides a better fit to the data. Would it be possible to include a model into the comparison in which PEs are also generated for 'pain absent' trials, so that these can also be formally compared? At least the authors should discuss this issue, as it can by no means be taken for granted that no PEs are generated for 'stimulus absent' trials.*

We agree that the question of how exactly pain prediction errors (PE) are represented in the brain is highly important. While reward prediction error encoding is well understood, the verdict on aversive PE is still out (Seymour et al., 2005; Belova et al., 2007; Bromberg-Martin et al., 2010; Fiorillo, 2013; Klavir et al., 2013; McHugh et al., 2014; Roy et al., 2014; Matsumoto et al., 2016; see below). Furthermore, appetitive and aversive PE seem to be encoded in different brain regions (Yacubian et al., 2006; Seymour et al., 2007) and different neuronal populations (Belova et al., 2007; Fiorillo, 2013).

From a practical point of view, neither signed PE nor absolute PE models (as suggested by this reviewer) offer a better fit in the current study than the ‘pain PE’. We now report the results of formal model comparisons of the different PE implementations in the main Results section (Figure 7). We also extended the Discussion to include these results and their implications.

In analogy to the above-mentioned studies on visual and auditory perception, an unexpected omission of pain (cue high, warm stimulus) results in more anterior insula activity than an expected omission (cue low, warm stimulus).

Furthermore, asymmetric or category specific PE have also been observed in face perception (den Ouden et al., 2010; Egner et al., 2010). In the study by den Ouden et al. (2010), fusiform face area activity was modulated by an unexpected face presentation, but not by unexpected house presentations (similar to our observations). In contrast, activity in the parahippocampal place area was modulated by unexpected presentations of both houses and faces, reflecting a signed PE. This suggests that prediction error coding might vary across modalities or perceptual categories, which is also suggested by distinct topographies of PE’s for warm and pain stimuli (Ploghaus et al., 2000; Zeidan et al., 2015).

*4) Subjective perception of pain:*

*Looking at the results, it is true that the SCR, pupil dilation and BOLD signal can be modeled using a predictive coding model similar to the one used by Egner et al. (2010). However, along the whole study, these results are not linked in any manner to the participant's subjective perception of pain. Previous studies investigating predictive coding showed that predicting sensory information enhances the representation of the predicted input and therefore its perception. When reading the manuscript, some immediate questions come to mind: Does predictive pain enhance perception of pain compared to unpredicted pain? What would an enhanced representation of pain imply? (more or less subjective pain?) When people successfully predict a painful stimulus, is the stimulus perceived more or less painful?*

These are very interesting questions. A recent study did observe no difference in pain ratings between correctly and incorrectly cued pain stimuli (Zeidan et al., 2015). Another study on visual perception observed a sharpening of stimulus representation by expectations (Kok et al., 2012). In pain perception, this could be reflected in better discriminability of slightly different stimuli, for example by measuring just-noticeable-differences or by assessing spatial discriminability of pain stimuli. In this study, we had to use a very time-efficient design and thus did not record pain intensity ratings for every trial to answer this question directly.

However, responses of the Neurological Pain Signature (NPS; a multivariate pattern of fMRI voxel-weights) have been shown to correlate well with perceived heat pain intensity (Wager et al., 2013; Krishnan et al., 2016) and can be used as a proxy for perceived pain here. This proxy did show a main effect of stimulus intensity (Table 1, Figure 3) and a trend towards an interaction (p=0.12; Table 1). Inspecting the parameter estimates and computing a post-hoc t-test (t(27)=2.2; p=0.036), shows that unexpected pain tends to elicit higher NPS responses compared to expected pain. This post-hoc finding would have to be confirmed in future studies that also acquire pain intensity ratings. Skin conductance and pupil dilation also correlate with pain perception (Geuter et al., 2014) and could also serve as a proxy here. Both measures show strong interaction effects, again with stronger responses to unexpected than expected pain.

Based on these proxies (NPS, pupil, SCR) we would expect perceived pain to be lower when it is expected than when it is not expected. Comparing the proportion of stimuli categorized as painful between the high and low probability cues revealed no significant difference in the current sample (mean_cuelow_=0.94; mean_cuehigh_=0.96; p=0.11; Z=1.6; Wilcoxon signed rank test). However, the power of this comparison is expected to be extremely low, because we calibrated the stimuli to be clearly dissociable.

Since we agree with the reviewer that this is an important field for future studies, we now mention this issue (as well as the post-hoc NPS results) in detail in the Discussion.

*5) Given the two alternatives design (painful Y/N) and that warm/painful stimuli were clearly dissociable, there may not be enough variance in the data in order to correlate subjective pain ratings with the changes in brain activity across experimental conditions. Showing the behavioral correlates of each condition may nevertheless improve the significance of the study.*

The reviewer is correct in stating that there is not enough variance in the rating data for a brain-behavior correlation. We now mention this explicitly in the manuscript: “Note that we did not carry out any correlational analyses between ratings and fMRI data, as the additional variance in the ratings (on whether a stimulus was perceived as warm or painful) is minimal when compared to the actual stimulation: only 5% of the responses did not match the stimuli and 18% of these mismatching responses were given on the first trial of an fMRI run.”